# Life tables in entomology: A discussion on tables' parameters and the importance of raw data

Luca Rossini[1]*, Mario Contarini[2], Stefano Speranza[2,3], Serhan Mermer[4,5], Vaughn Walton[4], Frédéric Francis[6], Emanuele Garone[1]

**1** Service d'Automatique et d'Analyse des Systèmes, Université Libre de Bruxelles, Brussels, Belgium, **2** Dipartimento di Scienze Agrarie e Forestali, Università degli Studi della Tuscia, Viterbo, Italy, **3** Centro de Estudios Parasitológicos y de Vectores (CEPAVE, CONICET-UNLP), La Plata, Argentina, **4** Department of Horticulture, Oregon State University, Corvallis, Oregon, United States of America, **5** Department of Environmental and Molecular Toxicology, Oregon State University, Corvallis, Oregon, United States of America, **6** Functional and Evolutionary Entomology, Gembloux Agro-Bio-Tech, University of Liège, Gembloux, Belgium

* luca.rossini@ulb.be

## Abstract

Life tables are one of the most common tools to describe the biology of insect species and their response to environmental conditions. Although the benefits of life tables are beyond question, we raise some doubts about the completeness of the information reported in life tables. To substantiate these doubts, we consider a case study (*Corcyra cephalonica*) for which the raw dataset is available. The data suggest that the Gaussian approximation of the development times which is implied by the average and standard error usually reported in life tables does not describe reliably the actual distribution of the data which can be misleading and hide interesting biological aspects. Furthermore, it can be risky when life table data are used to build models to predict the demographic changes of the population. The present study highlights this aspect by comparing the impulse response generated by the raw data and by its Gaussian approximation based on the mean and the standard error. The conclusions of this paper highlight: *i)* the importance of adding more information to life tables and, *ii)* the role of raw data to ensure the completeness of this kind of studies. Given the importance of raw data, we also point out the need for further developments of a standard in the community for sharing and analysing data of life tables experiments.

## 1. Introduction

Life tables are a powerful tool, widely used since the late 1960s in the entomology and ecology communities [1] to summarise the most important parameters of the life cycles of insects. They relate the development of the individuals of the population to the conditions of the surrounding environment [2] (e.g. temperature), highlighting the ectothermic nature of insects, or to the effect of different agents such as diet, pesticides, or natural enemies [3].

Life tables values are estimated by rearing a cohort of eggs (laid on the same day or on a shorter time) in climatic chambers at a single or at a range of biologically relevant constant temperatures [2,4]. Each individual of the cohort is monitored at a specific time frequency,

**Data Availability Statement:** All data utilized in this study, along with the accompanying scripts necessary for complete result reproduction, are

openly accessible via the following link: https://github.com/lucaros1190/LifeTablesIssues.

**Funding:** LR is funded by the European Commission under the Grant n. 101102281, Project "PestFinder", call HORIZON-MSCA-2022-PF-01. Part of this work has been supported by the Fons de la Recherche Scientifique-FNRS under the Grant n. 40003443 ("Smart Testing") and by the Brussels Institute of Advanced Studies (Grant BrIAS2024). The funders had no role in study design, data collection and analysis, decision to publish, or preparation of the manuscript.

**Competing interests:** The authors have declared that no competing interests exist.

usually one day, when its current life stage is recorded [2]. The inspections are repeated until the death of the individuals from a population [3,5,6], and the data are usually reported in specific matrixes as described by Chi and Liu [7], Chi [5], and Chi et al. [3,6]. These experiments allow researchers to obtain various information such as the age-stage distribution of all individuals over time and, by changing the environmental conditions of the growth chamber, as a function of temperature, relative humidity, and photoperiod [4,8,9]. The age-stage distribution describes the duration, usually expressed in days, of the biological stages that compose the insects' life cycle: such information is extremely helpful to analyse the bioecology of the insect species and to formulate mathematical models that describe their biology.

Once these experimental data are obtained, life tables are constructed by reporting, for each rearing temperature, synthetic information extracted from the raw data such as: the mean development time of each life stage, the mean number of eggs laid per female and per day, and the mean survival time for all life stages. These mean values are usually reported with the associated standard deviation or, more commonly, their standard error. Additional information that is sometimes reported in life tables studies includes the net reproduction rate [10,11], i.e., the mean number of individuals that may be born from a female at given environmental conditions, and the mean generation time [10], i.e., the mean duration between the birth of an individual and the birth of an individual of the next generation. Mathematically, the net reproduction rate $R_0$ is defined as

$$R_0 = \sum_{x=1}^{k} l_x m_x \tag{1}$$

where $x$ is the age, $k$ is the maximum age for each stage, $l_x$ is the age-specific survival rate, and $m_x$ is the age-specific fecundity [3]. Given this definition, the mean generation time $T_G$ can be written as follows:

$$T_G = \frac{1}{R_0} \sum_{x=1}^{k} x l_x m_x. \tag{2}$$

Life tables have a well-defined mathematical theory to analyse the biological traits of the population, but it is efficient only in case of fixed environmental conditions [3,5,7]. Previous studies (e.g., [4,10,12–16]) indicated that life tables can also be used to underline the ectothermic nature of insects that is also very interesting for pest management and population modelling purposes. Indeed, insect development occurs only within a certain thermal range. The insect development is theoretically not possible outside such thermal range. Starting from the lowest temperature threshold, the mean development time of the population decreases until reaching a minimum, coinciding with the optimal temperature for the species development. After this minimum, the development time starts to increase again until reaching the maximum temperature threshold. This particular profile aroused the interest of several authors who proposed various mathematical functions to interpolate these life tables' trends [17–25]. A similar dependency on temperature can be observed for mortality incidence and fecundity rates [26–28].

The most common approach to study this profile is the following [4,29]: the mean development times $D_i(T)$ of each stage $i$ are converted into *development rates* $G_i(T)$ described by the equation

$$G_i(T) = \frac{1}{D_i(T)}. \tag{3}$$

To use development rates $G_i(T)$ instead of mean development times $D_i(T)$ has the advantage of transforming the decreasing-increasing profile of the development times into an increasing-decreasing one, with a maximum instead of a minimum. Moreover, as temperature approaches the thermal limits, the development time increases up to infinity, while under the same conditions the development rates are more practical, as they approach zero. The profile tracked by the development rates is usually interpolated using various mathematical functions [17,19,20,25], commonly called "*development rate functions*" [9,21]. It is commonly accepted in the literature that, once the parameters of a given development rate function are estimated, they are species-specific [9], and that if deviations from these rates are experienced, they might be due to the effect of genetic mutations or environmental adaptations.

Although life tables have been "standardised" by the scientific community over several decades [3,5–7], in this paper we raise some concerns about the completeness of the information they contain. Our starting point is the observation that summarising the actual distribution of the development times using only averages and standard errors implicitly assumes that the actual form of the distribution is not relevant.

However, from a practical viewpoint it is widely understood that the shape of the distribution contains important information to describe the biology of insects. For instance, the shape of the distribution (e.g. the presence of multiple peaks) can give important information on the genetic variability of the species and might be linked to recent studies investigating intraspecific genetic variability and adaptation potential (e.g., [30]).

The actual shape of the distribution is also very useful for pest control purposes: for instance the minimum and maximum development times are fundamental to avoid useless treatments in planning pest control actions, while the shape and the range of the distribution provides an idea of the capability of the species to survive through anomalous adverse events. Indeed, the larger the distribution of development times the larger the resilience of the species against punctual anomalous events. When contextualised in terms of pest control, this aspect significantly influences the choice of the control method.

Clearly, the fact that life tables only report mean values and standard errors to describe development times is not a limiting factor whenever authors provide raw datasets on the experiments as supplementary material of their publications. However, in the entomological community, this is not yet common practice, which produces a loss of possibly precious information for the scientific community. Remarkably, the unavailability of data does not allow third parties to verify and possibly refine the provided data analysis, which is a foundation of the scientific method.

In this paper we do not raise any doubt on the rigour and validity of experimental protocols, which in our opinion represent state-of-the-art scientific methods, nor the usefulness of life tables as a tool to summarise information, but the fact that the current practice of our community of presenting only summarised data may produce a loss of relevant information, and possibly lead to erroneous conclusions. Different authors over the years overcame the issue related to the distribution by analysing the data through the bootstrap method [31], however this technique implicitly assumes that the individuals of the population are interchangeable to each other. Accordingly, we miss the information on the actual distribution of the development times, a relevant characteristic of each species.

This critical study also underlines the possible negative impact that the sole use of synthetic data in life tables has on biological information and modelling of insect populations and points out the need for further developments of a standard among entomologists for sharing the data collected in life tables experiments. This paper targets both entomologists involved in biological studies and researchers focusing on the construction of insect population models, as it demonstrates that the exclusive usage of life tables' synthetic information might hide essential biological information and might lead to pest models that are not sufficiently predictive.

## 2. Materials and methods

### 2.1. Presentation of the problem

In order to define and explain the problem, let us consider the concrete case of a cohort of eggs developing at fixed constant temperatures. In particular, we consider the development of the eggs of the rice moth *Corcyra cephalonica* (Stainton) (Lepidoptera: Tortricidae), whose life tables data have been published by Rossini et al. [32] not accounting for reproduction, and for which the raw dataset is available at https://github.com/lucaros1190/DatasetCorcyra-cephalonica. The life cycle of this species is composed of an egg stage, six larval instars, a pupa stage, and adult males and females [33,34].

Based on the guidelines usually accepted in the literature [4], the experiment of Rossini et al. [32] was carried out as follows: *i)* cohorts of eggs provided by a continuous rearing of the species were placed in thermal-controlled growth chambers at different constant temperatures, and *ii)* the development of each individual has been followed for the whole life cycle [5]. The temperatures explored were 18, 21, 24, 26, 28, 30, 34, and 36˚C.

Let us focus, for the sake of simplicity, on some representative temperatures (21, 26, 28, and 30˚C) and only on the development from the egg stage to the adult emergence, namely: egg, larvae (with no distinction between larval instars), and pupae. To better identify the problem, we focus only on the survived individuals, with no consideration on mortality. This fact does not affect the rationale of this study, because the stage-development time in life tables is calculated based on the individuals that survive the whole stage of interest [12,32,35–37].

The values listed on life tables are usually summarised by the average of the development time of each individual of the population and its standard error. An example for the species at hand is reported in Table 1 and reports the average and the standard error of the development time of the egg, larvae, and pupa stages for the four constant temperatures (21, 26, 28, and 30˚C).

The main insight behind the current study is that egg hatching and stage development have characteristic time-distributions around the peak [5,12,38] which in most cases cannot be reconstructed by using only average and standard. Indeed, the only relevant case where reporting the average and the standard error would be without loss of information is the case where the development times follow a Gaussian distribution. However, as we argue in this

**Table 1. Life tables parameters of *C. cephalonica* at constant temperatures (21, 26, 28, and 30˚C).**

| Life stage | Temperature (˚C) | Mean development time (days) | Number of individuals (n) |
|---|---|---|---|
| Egg | 21 | 8.0±0.1 | 32 |
| | 26 | 4.8±0.1 | 136 |
| | 28 | 5.0±0.0 | 62 |
| | 30 | 4.2±0.1 | 129 |
| Larva | 21 | 44.5±0.9 | 32 |
| | 26 | 24.4±0.4 | 136 |
| | 28 | 22.7±0.3 | 62 |
| | 30 | 23.6±0.5 | 129 |
| Pupa | 21 | 21.7±0.4 | 32 |
| | 26 | 14.5±0.3 | 136 |
| | 28 | 13.0±0.3 | 62 |
| | 30 | 11.3±0.2 | 129 |

Average development times (mean ± standard error) and initial number of individuals from Rossini et al. [32]. The dataset is related only to *Corcyra cephalonica* egg, larvae, and pupa stages.

paper, this assumption is often incorrect and highly depends on the specific dataset. Let us take as an example the plot in Fig 1, where we overlapped the experimental distribution of the development times and the Gaussian distribution centred in its average value and width given by its standard deviation. As we can see, there is a notable shift between the peaks, which may lead to subsequent wrong interpretation of the biology of the species. For instance, this can be a problem when life table values are used to deduce the mean generation time of the population under given environmental conditions, which is fundamental during the planning of pest management strategies.

A more in-depth understanding of the distribution of development times specific to life stages and temperatures can significantly enhance the amount of biological information. The distribution of the times serves as macroscopic evidence of the physiological and biochemical processes underlying development, which, in turn, depend on living conditions. An additional consequence of an incorrect synthesis of the raw data is illustrated in Fig 1, where an apparently bimodal distribution of the data is approximated with a Gaussian curve.

Multimodality is a phenomenon that frequently occurs in biological datasets [39–43] and it implies that individuals from a single population can have two different developmental times under the same environmental conditions. A typical example is represented by the European chestnut weevil, *Curculio elephas* (Gyllenhal), where a portion of the larvae become adults in the same season, and the other after one or more years [44]. Utilizing mean and standard error to synthesise the dataset in Fig 1 implies a substantial loss of information that could be valuable instead for exploring different aspects of species development. From a pest control perspective, for instance, a Gaussian approximation of bimodal datasets means neglecting the two peaks of

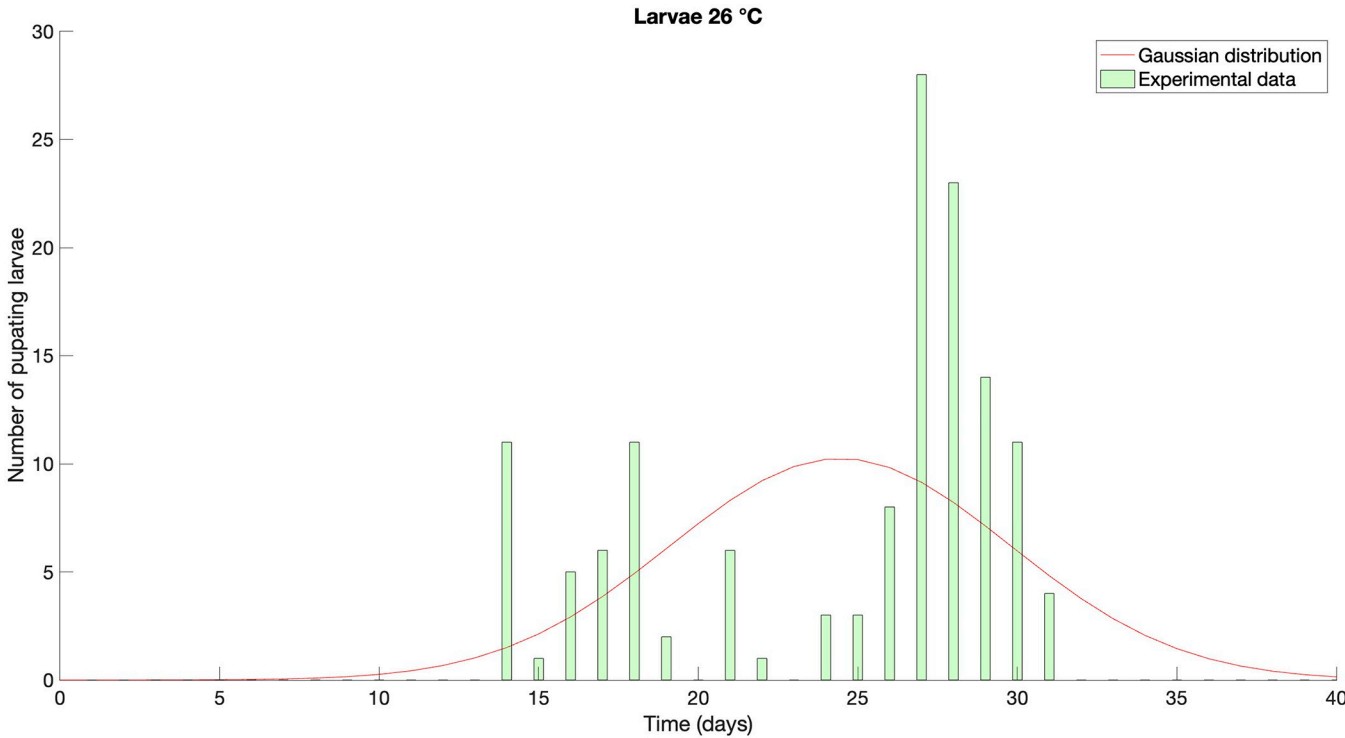

**Fig 1. Gaussian distribution obtained by the average development time and the standard deviation of the experimental data [32] (red line), *versus* the raw experimental dataset.** The plot shows data specifically for the larval phase at constant temperature of 26˚C, as an example of how the peak and the range indicated by the Gaussian distribution fairly differs from the real peak(s) of the population.

the distribution, resulting in overestimation and underestimation, respectively, of the time of action in case of the first and the second peak.

Other possible issues arise when life tables are used for pest population models. Since this is an issue common to several pest population models available in the current state-of-the-art, a more detailed explanation follows in the next section.

## 2.2. Life tables data and models

In recent years, the increasing demand of Decision Support System tools for the management of insect pest species is pushing the development of more reliable insect population models [45–52]. Life tables play a fundamental role in the development of these models. Indeed, the most popular models describe insect population dynamics as a population developing over time and through their life stages, see e.g. [53–63] and are characterised by parameters such as e.g. stage development, fecundity, and mortality, which are dependent on environmental parameters, mainly temperature [13,15,27,62,64]. In most cases, these parameters are set on the basis of the synthetic data reported in life tables studies, leading to the conclusion that better models could be devised if the raw data of these experiments were available.

To understand what we mean, we recall here that the standard protocol to construct life tables consists in placing a cohort of eggs, laid on the same day, in a climatic chamber. Then, the stage of each individual is monitored on a daily (or hourly, depending on the species) basis. The dataset resulting from this kind of experiments provides the distribution of the times the insects spend passing from a given stage to the next one (e.g., from eggs to the first larval instar), on the basis of which the various development/mortality/fecundity rates are computed according to the standard conventions. Interestingly, if we think about this protocol from the point of view of the system identification (i.e., the formulation of mathematical models based on measurements and observations of a natural phenomenon) and if the cohort of eggs is sufficiently large, it is immediate to realise that this experiment can be seen as an "impulse response" identification experiment for a discrete time system [65]. The choice of the discrete time is driven by the experimental settings that, as already mentioned, usually provide for daily (and accordingly discrete) inspections.

To clarify what we mean, let us define the input $u_e(t)$ as the number of new eggs laid at time $t$, and as output $y_{l_1}(t)$ the number of larvae hatching at time $t$. The protocol puts at an initial time $t = 0$ a certain number $N$ of eggs in the system, which is equivalent to say that $u_e(t) = N\delta(t)$ where $\delta(t)$ is the Kroeneker impulse, i.e.

$$\delta(t) = \begin{cases} 1 & if \quad t = 0 \\ 0 & if \quad t \neq 0 \end{cases}$$

The recorded sequence of hatching $\{y_{l_1}(0), y_{l_1}(1), \ldots, y_{l_1}(T), 0, 0, \ldots\}$ is the response for an impulse of amplitude $N$ which allows to write that, on the basis of the data collected in this kind of experiments, the impulse response of the system is the sequence

$$w_{e,l}(t) = \left\{ \frac{y_{l_1}(0)}{N}, \frac{y_{l_1}(1)}{N}, \ldots, \frac{y_{l_1}(T)}{N}, 0, 0, \ldots \right\} \tag{4}$$

The interesting aspect of impulse responses is that for linear time-invariant systems they are themselves the input/output model of the system. In particular, given the impulse response

$w_{l_1,e}(t)$, the output is the sum of convolution between the input and the output response:

$$y_l(t) = \sum_{\tau=-\infty}^{t} w_{l,e}(t-\tau)u_e(\tau).$$

As well known, this representation can also be rewritten in the $Z$-transform domain as

$$Y_l(z) = G_{l,e}(z)U_e(z) \tag{5}$$

where $G_{l,e}(z) = Z\{w_{l,e}(t)\}$ is the so-called transfer function of the system.

By repeating the procedure for the transition from each insect's life stage to the next one (including from each stage to death) we can obtain a set of transfer functions representing the sub-model from each development stage to the next. It is thus possible to build a complete model for the dynamics for the single stages or for portions of the life cycle by composing all these transfer functions.

Following this procedure for the specific dataset of *C. cephalonica*, we can thus compute the impulse responses from eggs to larvae, $w_{e,l}(t)$, from larvae to pupae $w_{l,p}(t)$, and from pupae to adults, $w_{p,a}(t)$ for each of the four constant temperatures considered in the example. Note that for eggs all the individuals start from the day zero, while for larvae and pupae the dataset needs a reorganisation to "shift" the population of the alive individuals to time zero for every stage. After the dataset is correctly organised, we can compute the transfer function (5) of each impulse response considering the overall portion of survived individuals, $S_i \in [0, 1]$, as well, i.e.

$$Y_l(z) = S_e G_{l,e}(z)U_e(z) \tag{6}$$

$$Y_p(z) = S_l G_{p,l}(z)U_l(z)$$

$$Y_a(z) = S_p G_{a,p}(z)U_p(z)$$

were the subscript $l$, $p$, and $a$ describe the egg hatching, pupating larvae, and emerging adult phases, respectively. Note that most of the life tables report the initial and final number of individuals for each experiment, from which it is possible to calculate the survival rate $S_i$.

By composing these three transfer functions it is also possible to get the transfer function between eggs and each other stage

$$Y_l(z) = S_e G_{l,e}(z)U_e(z) \tag{7}$$

$$Y_p(z) = S_e S_l G_{p,e}(z)U_e(z)$$

$$Y_a(z) = S_e S_l S_p G_{a,e}(z)U_e(z)$$

where $G_{p,e}(z) = G_{p,l}(z)G_{l,e}(z)$ and $G_{a,e}(z) = G_{a,p}(z)G_{p,l}(z)G_{l,e}(z)$.

Of course, the possibility to build these models relies on the availability of life tables raw dataset, justifying what we stated in Section 2.1, i.e., the importance of having access to the experimental datasets.

## 2.3. Methodology

To demonstrate the limits of the synthetic data in describing the insects' life cycle we considered two aspects: *i)* the difference between the actual development time distribution and the Gaussian approximation resulting from the life-table synthetic data (Fig 1); *ii)* the difference

between prediction models based on the raw dataset and the ones resulting using the life-table synthetic data.

**2.3.1. Differences between the real distributions of the development times and their Gaussian approximation.** We compared the real distributions of the development times for each life stage and temperature with the Gaussian approximation reported in the life tables. We recall that the Gauss distribution is defined as

$$f(t) = \frac{1}{\sigma_D \sqrt{2\pi}} e^{-\frac{(t-\mu_D)^2}{2\sigma_D^2}} \tag{8}$$

The two parameters characterising any Gaussian distribution are the mean value, $\mu_D$, and the standard deviation, $\sigma_D$, which can be obtained by the definition of standard error, $St_{err}$ as

$$\sigma_D = St_{err} \cdot \sqrt{N} \tag{9}$$

where $N$ is the number of specimens reared.

To assess similarity and discrepancies, we compared the real distribution and the Gaussian distribution resulting from the life tables both graphically, by plotting the dataset and the Gaussian curve, and quantitatively, by performing for each life stage and temperature a test of Shapiro-Wilk (S-W) with a threshold $p$-value of 0.05. We recall that in the Shapiro-Wilk test datasets with $p<0.05$ are considered not normally distributed [66]. The results of the Shapiro-Wilk test were further compared by visual inspection of quantile-quantile (Q-Q) plots, reported as S1 Fig.

For each experimental dataset we also computed:

- The median value, namely the middle value separating the greater and lesser halves of the dataset;

- The mode, namely the most frequent value of the dataset;

- The kurtosis [67]

$$k = \frac{\frac{1}{N}\sum_{i=1}^{N}\left(x_i - \mu_D\right)^4}{\left[\frac{1}{N}\sum_{i=1}^{N}\left(x_i - \mu_D\right)^2\right]^2} \tag{10}$$

namely the thickness of the tails of the distribution;

- The skewness [67]

$$s = \frac{\frac{1}{N}\sum_{i=1}^{N}\left(x_i - \mu_D\right)^3}{\left[\frac{1}{N}\sum_{i=1}^{N}\left(x_i - \mu_D\right)^2\right]^{3/2}} \tag{11}$$

namely the measure of the asymmetry of the distribution.

It is worth reminding that in the case of a Gaussian distribution mode, median, and mean assume the same value, while kurtosis is 3 and skewness is zero [68], and that deviations from these nominal values are further indicators of how far the data distribution is from the Gaussian. Additionally, kurtosis serves as an index indicating how "flat or sharp" the peak of a distribution is in comparison to a normal distribution (k>0 for a sharp peak, k<0 for a flat peak), while the skewness acts as an index of the distribution's symmetry (s>0 for right-skewed, s<0 for left-skewed).

**2.3.2. Differences between the population dynamics modelled considering the impulse responses from real data and from their Gaussian approximation.** We hereby compared

the population dynamics described by models (7) considering the impulse responses calculated by the life tables raw dataset and by its Gaussian approximation (8). For the sake of a better visualisation all the populations will be normalised to avoid having to consider mortality. The purpose of this part of the study is to show how the differences highlighted in Section 2.3.1 are further amplified if the evolution of the population is predicted by models.

**2.3.3. Data availability and analysis.** The raw data, as well as the scripts to fully reproduce the result of this study are both publicly available at https://github.com/lucaros1190/LifeTablesIssues or provided as Supporting information of this paper (S1 File). For the sake of completeness, the shared scripts extend the analysis to all the eight temperatures considered in Rossini et al. [32]. The analysis of the impulse response was carried out in Matlab (vers. R2023a, MathWorks, USA) using the *tf* function to calculate the transfer function and *impulse* for the impulse response, while the Shapiro-Wilk test was carried out using the *shapiro.test()* function within R Studio (vers. 4.2.3, R Core Team). We refer the most interested readers to the shared scripts (link or S1 File) for the full list of software packages and functions involved in the calculations.

# 3. Results and discussion

## 3.1. Differences between real distributions of the development times and their Gaussian approximation

In this subsection we report the difference between the actual experimental distributions and the ones obtained by considering the Gaussian distributions of the life tables' synthetic data (Fig 2 and S1 Fig). Furthermore, the *p*-values of the Shapiro-Wilk normality test were reported in Table 2, as well as the additional parameters of descriptive statistics mentioned in Section 2.2. From an inspection of the figures and of the numbers, and from the Q-Q plots provided in S1 Fig, it is apparent that in almost all the cases it is not reasonable to claim that the experimental distributions are normally distributed.

Indeed, the first observation is that for the passage between egg and larvae there is a qualitatively good overlap between the raw dataset and the Gaussian distribution (Fig 2, first column of plots), even if the Shapiro-Wilk test and the Q-Q plot indicates the non-normality of the data for all the temperatures (Table 2 and S1 Fig). As the life cycle advances, some major discrepancies become more apparent in the plots as well. A clear example is the distribution of the development times of larvae at 21°C (Fig 2, second column), where the raw dataset is heavily skewed (skewness 1.31, Table 2) and where there is a time shift of 10 days, *circa*, between the peak of the real distribution and of the Gaussian, and of 2 days between the peak of the Gaussian and the median. Analogous situations, even if with a lower time shift, can be observed in the case of larvae and pupae at 28°C (median 22, mode 20, skewness 0.89, Table 2), and by pupae at 30°C (median and mode 11, skewness 0.17, Table 2). Finally note that the Shapiro-Wilk test and the Q-Q plots in S1 Fig indicate the non-normality of the datasets for all cases except that for the pupae at 21°C.

These simple observations clearly point out that the synthetic data reported in the current life tables come with a loss of information on the form of the distribution. In line of principle, different solutions to overcome this problem could be devised, such as increasing the number of computed parameters (e.g., adding some of the parameters reported in Table 2) or using other distributions and their characteristic parameters. This observation echoes the intuition of Wagner et al. [8], which was probably the first to mention that the distribution of the development times should go beyond the Gaussian approximation. For the sake of completeness, it is worth mentioning here that over the years some alternatives to the normal distribution have been proposed. Indeed, in the same paper, Wagner et al. [8] proposed to approximate the

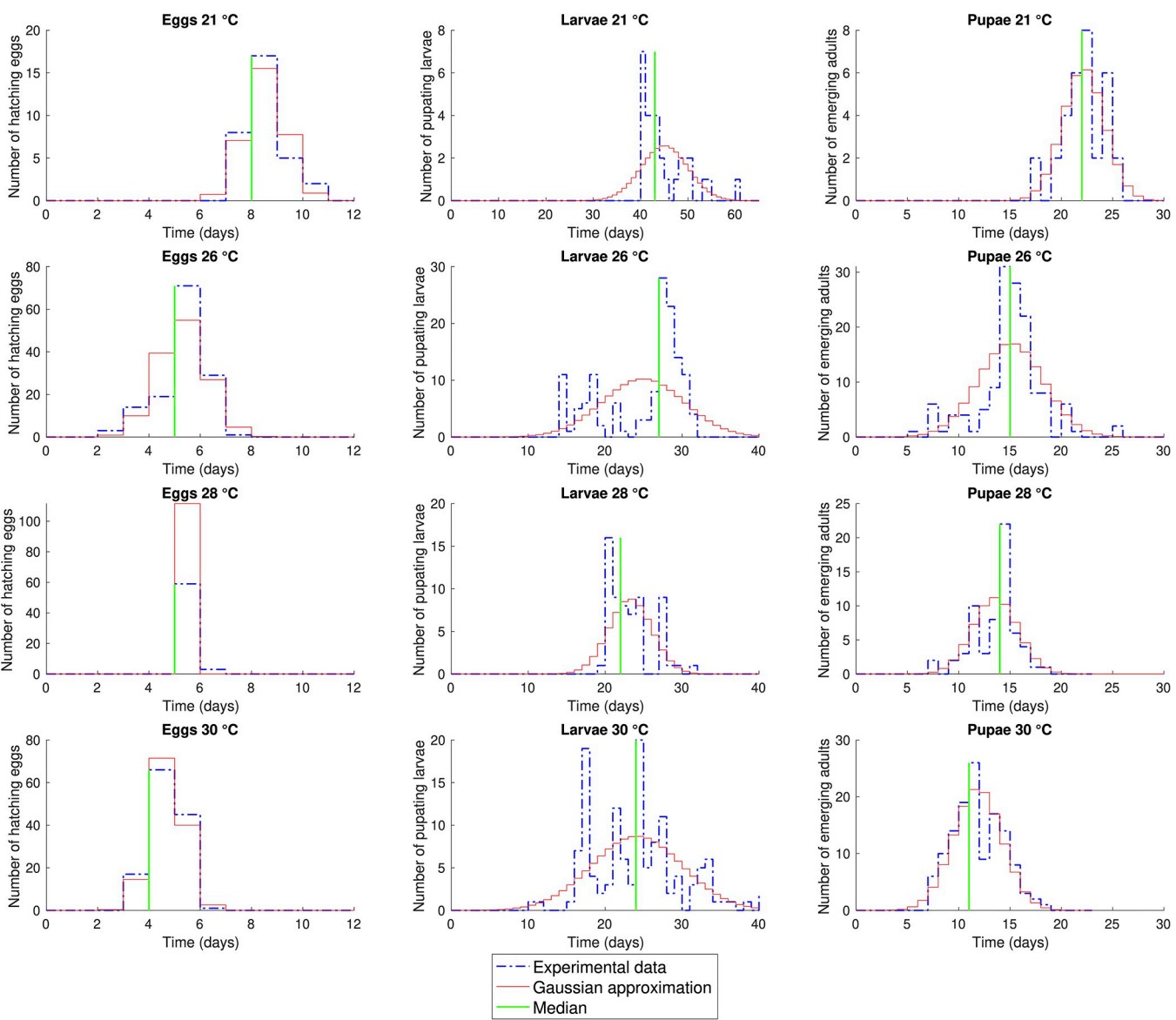

**Fig 2. Distribution of the single stages at 21, 26, 28 and 30°C calculated by the experimental data and by the Gaussian distribution obtained by mean and standard error of the development times of *Corcyra cephalonica* individuals.** The green line indicates the median of the experimental data. This figure is also an example of the difference between a model built using the Gaussian hypothesis and the model built using the actual distribution of the data.

frequency distribution of the development times with a Weibull function. Poisson or negative continuous binomial distributions were proposed [69] for continuous-time distribution, and the Erlang distribution was proposed [70–72] for discrete time. Some authors have instead suggested the use of Bayesian approaches [73], to circumvent the distribution of the data, but these methods encounter challenges when dealing with multimodality.

However, it is important to underline that the wide adoption of any specific distribution would require a large amount of data on a large number of different species for its validation and wide acceptance which, given the very small amount of publicly available data, is currently not realistic. Furthermore, any synthetic parameters from descriptive statistics will always come with some loss of information and an extra burden for the analysis of the results. Any

**Table 2. Additional parameters calculated from the raw dataset of Rossini et al. [32] listed in Table 1.**

| Life stage | Temperature (˚C) | Mean development time (days) | Mode (days) | Median (days) | Kurtosis | Skewness | S-W test p-value* |
|---|---|---|---|---|---|---|---|
| Egg | 21 | 8.0±0.1 | 8 | 8 | 3.16 | 0.65 | $1.7 \cdot 10^{-4}$<br>$W = 0.83$ |
| | 26 | 4.8±0.1 | 5 | 5 | 3.46 | −0.81 | $1.6 \cdot 10^{-10}$<br>$W = 0.85$ |
| | 28 | 5.0±0.0 | 5 | 5 | 18.71 | 4.21 | $2.2 \cdot 10^{-16}$<br>$W = 0.22$ |
| | 30 | 4.2±0.1 | 4 | 4 | 2.39 | −0.17 | $1.0 \cdot 10^{-11}$<br>$W = 0.81$ |
| Larva | 21 | 44.5±0.9 | 40 | 43 | 4.27 | 1.31 | $2.5 \cdot 10^{-4}$<br>$W = 0.84$ |
| | 26 | 24.4±0.4 | 27 | 27 | 2.13 | −0.81 | $3.0 \cdot 10^{-11}$<br>$W = 0.83$ |
| | 28 | 22.7±0.3 | 20 | 22 | 2.96 | 0.89 | $2.0 \cdot 10^{-5}$<br>$W = 0.88$ |
| | 30 | 23.6±0.5 | 24 | 24 | 2.92 | 0.43 | $2.0 \cdot 10^{-3}$<br>$W = 0.96$ |
| Pupa | 21 | 21.7±0.4 | 22 | 22 | 2.84 | −0.43 | **0.10**<br>$W = 0.94$ |
| | 26 | 14.5±0.3 | 14 | 15 | 4.81 | −0.23 | $7.1 \cdot 10^{-7}$<br>$W = 0.92$ |
| | 28 | 13.0±0.3 | 14 | 14 | 3.45 | −0.62 | $2.0 \cdot 10^{-3}$<br>$W = 0.93$ |
| | 30 | 11.3±0.2 | 11 | 11 | 2.31 | 0.17 | $3.0 \cdot 10^{-3}$<br>$W = 0.97$ |

This set of parameters should be applied as standard of life tables studies to report results.

* S-W means Shapiro-Wilk normality test, as described in Section 2.3.1. A *p*-value less than 0.05 means that the dataset does not follow a Gaussian distribution. Additional information to support the results listed in the table is the list of Q-Q plots reported as S1 Fig. For convenience, the normally distributed datasets are highlighted in bold.

assumption and choice, including how the sampling size affects the shape of the distribution, could be the object of further discussions/doubts in the future. For the moment, we can say that the case study considered in this work leaves to suppose that the shape of the distribution might be related to the biology of the species and not only to the size of the sample. The datasets, in fact, had a conspicuous size even in case of temperatures close to the thermal limits, where usually reaching high numbers is difficult because of the high mortality rate and the long development times that significantly extend the duration of the experiments.

We believe that the most desirable approach to maximise the pool of information while minimising the efforts would be to systematically complement the life tables with plots of the experimental distributions as in Fig 2 and by publishing the raw data of the experiment as supplementary material attached to the paper. This would complete the representation and the theory of Chi and Liu [7] that, over the years, has been adopted by different authors but that, at the same time, does not consider the life history of each single individual to obtain the distribution of the development times. It is worth saying, in addition, that from the distribution of the development times of the single individuals it is possible to obtain the matrices of Chi and Liu [7], but not *vice versa*.

To reinforce this proposal, please note that reporting the frequency distribution of the experimental data based on the life traits of each individual is not new, and that the literature provides many examples where this is done in the fields of biology and ecology. For instance, we can look at how the length of human pregnancy is reported, a case where data are

conceptually similar to the insect's stage development analysed in life tables. The common way to report these data is by plotting the frequency distribution of the gestation times, see e.g. [74], while the synthetic representation is carried out considering the mean value, the standard error, the median, and the percentiles. A similar example can be found in [75], where instead of the frequency distribution the authors reported its cumulative probability, while the data are synthesised through the median and the inter-quartile range.

Conceptually similar approaches can be found also in zoology, see for instance the works of Nogalski and Piwczynsky [76] and Brakel et al. [77] on the gestation lengths of cattle. In this case, authors also plotted the frequency distribution of times as a result, summarising them in tables reporting the mean, the standard error, and the skewness.

To the best of our knowledge, there are only a handful of works in entomology and nematology reporting data in a similar way. For instance, a cumulative proportion of the egg hatching was reported by Young et al. [78] on nematode populations reared at different constant temperatures, while the distribution of the development times as in Fig 2 was reported by Severini et al. [71] and Yaro et al. [79]. It is worth remarking, however, that none of these works reports the raw experimental dataset as supplementary material or in repositories of public access. The stage frequency matrices proposed by Chi and Liu [7], that aimed to increase the quantitative information usually reported in the life tables studies, is of course an interesting way of representing the data. The stage frequency matrix is in fact almost comparable to the raw data, as showed for example by Severini et al. [71], where the graphical representation of the distribution of the development times was supported by the stage frequency matrix of eggs and larvae, and Candy [80] for the *Chrisophtarta bimaculata* (oliver) (Coleoptera: Chrysomelidae). As already stated, however, the matrices of Chi and Liu [7] are usually built by considering how many individuals are, at each sampling time, in a given stage, losing information on the life history of each single specimen. Tracking the age-stage distribution of each single individual is however a fundamental component, for instance, to investigate in depth the intra-population genetic variation from a modelling point of view.

We believe that the systematic availability of raw data and the plots of the distributions as in Fig 2 provides information of fundamental importance in applied entomology and pest control, i.e., the fields where life tables studies find the main application [81]. As a further example of the importance of having these data available, we can notice in Fig 2 that there is always a minimum delay that is respected in each stage of development. Let us take as an example the egg stage. It is known from the literature [32–34] that the optimum temperature for the development of *C. cephalonica* is around 26–28°C. Fig 2 shows that even in the optimal conditions of growth, no eggs hatch before at least 2 days. Furthermore, this minimum delay changes over temperature, and this information is usually missing in the classical synthetic life tables representation. The minimum delay of each life stage provides relevant information on the biology of the species such as the minimum amount of time where no development (e.g., egg hatching, adult emergence) is expected for each given constant temperature. This aspect has a direct application in pest control strategy, for instance, since it can be of great support to estimate empirically the occurrence of the time one should wait to observe the life stages most susceptible to a particular control method [82].

The example of *C. cephalonica* suggests another interesting aspect that motivates the need of publishing the life tables raw data and of plots such as Fig 2. In fact, the environmental conditions influence not only the expected value of the development time, but it may also change its variance. From a comparison between plots (Fig 2), it seems to appear that as the temperature goes far from the optimum value, the development times of the individuals are spread over a higher range. Biologically speaking this fact is reasonable, since the higher the deviation from the optimal temperature value, the higher is the probability that extreme adverse events

happen. Thus, a wider time spread of the individuals can be an advantage, because in the case where part of the population in a given stage dies because of extreme external conditions, there is still a portion of individuals in the previous or further stages that can ensure the species survival [83]. The systematic availability of plots (such as in Fig 2) and raw data would allow more in-depth studies on these aspects.

The final noteworthy aspect provided by the example of *C. cephalonica* is the occurrence of bimodal (or multimodal, more generally) distributions, as illustrated by the dataset of larvae at 26°C (Fig 2). Managing this type of dataset is often challenging and many solutions proposed in the existing literature involve data transformation [84–86]. While transforming the dataset can facilitate the application of statistical methods such as the Analysis of Variance (ANOVA), it may not be entirely suitable for extracting information such as minimum/maximum development time, for instance. Additionally, finding a transformation law that effectively normalises the data can be difficult in some cases, as well as assigning a biological meaning to the transformed variables. The use of the impulse response faithfully reproduces the dataset, overcoming all the aforementioned issues.

## 3.2. Differences between the population dynamics modelled considering the impulse responses from real data and from their Gaussian approximation

In this section, we will show that, if the Gaussian distribution implicitly suggested by the data reported in life tables is used to build population models, it can lead to models whose time evolution does not reflect the actual behaviour of the population, and thus have very low predictivity. This is a quite relevant negative side effect of the discussion carried out in Section 3.1 as models are becoming more and more important for the future of precision agriculture (e.g., to decide when to carry out insect pest control) as well as for process control in insect farming [52,87]. To make our point, we compared the results of a model obtained considering as an impulse response the Gaussian distribution deducted from the life tables, and the one obtained directly by the experimental dataset.

At 21°C (Fig 3) the peak of the Gaussian distribution of development times of the larvae occurred on day 53, while the impulse response model indicates day 48. The real data were mostly concentrated in the left side of the plot, (days 47–51), with a peak on day 50. The Gaussian distribution in this case strongly anticipated the emergence of the first larvae, ten days before the real case. The pupae showed a similar scenario, with a peak of the experimental data on day 72, coinciding with the impulse response description, while the Gaussian distribution predicts day 75.

At 26°C (Fig 4) the situation was the opposite since the distribution of the larvae and pupae was mostly concentrated on the right side of the plot. Larvae had two peaks: one on day 21 and one on day 33, both closely represented by the impulse response model. The Gaussian distribution indicated a peak on day 30, strongly underestimating the emergence time of the greatest part of the population. The pupae showed the same pattern, well described by the impulse response model and strongly anticipated by the Gaussian representation.

In the cases of 28 and 30°C (Figs 5 and 6, respectively) the expected value of the Gaussian distribution was overall shifted with respect to the real distribution, overestimating or underestimating the peak of the population, respectively. If we take into account the average values listed in Table 1, accordingly, we are anticipating or postponing the mean generation time. This value is very important for the definition of pest control strategies, because empirically speaking is the most common value that farmers and technicians consider while planning treatments.

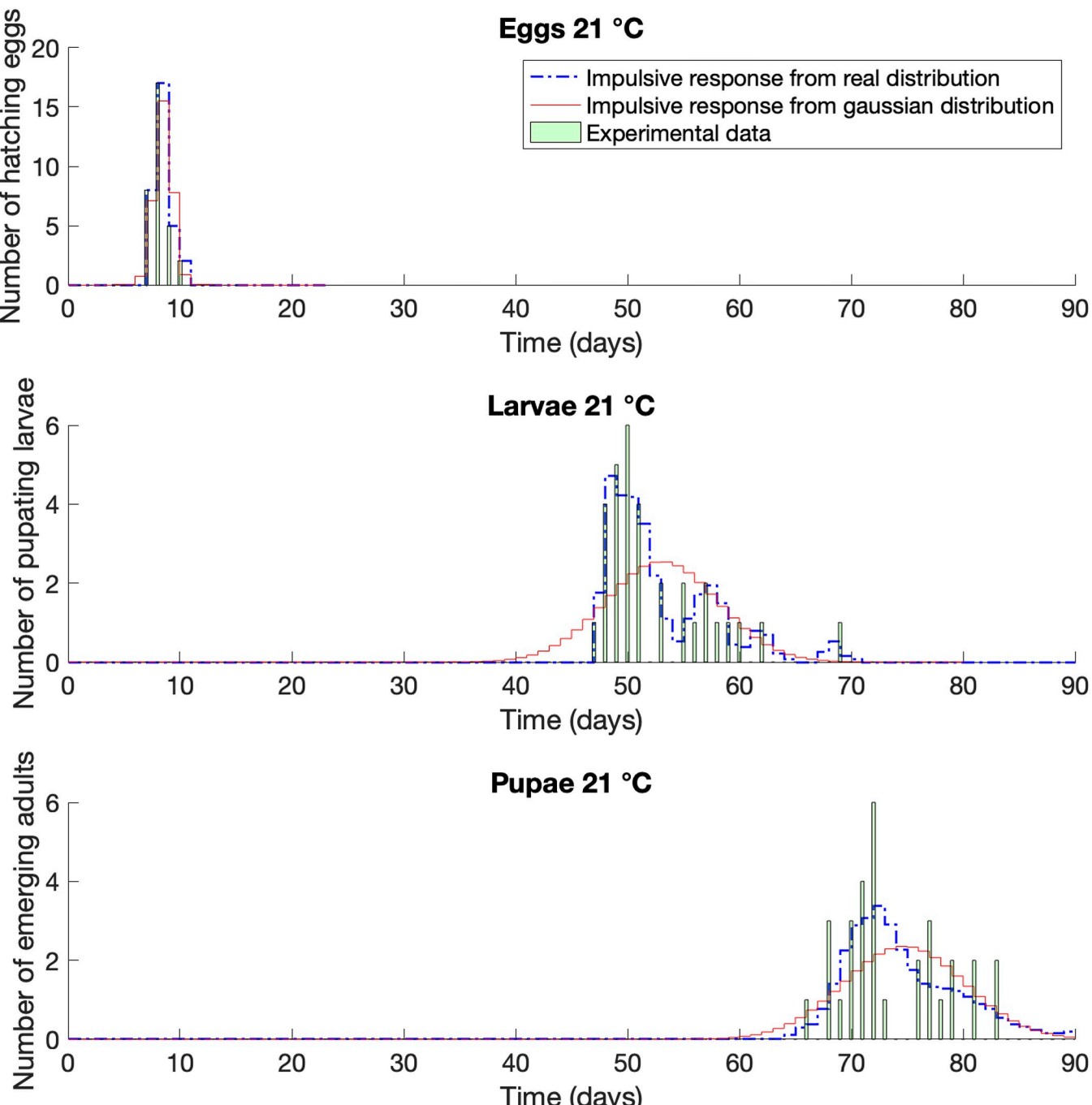

**Fig 3. Impulse response compared with life tables raw dataset** [32]**.** Case of *Corcyra cephalonica* at constant temperature of 21˚C.

## 4. Conclusion

Although life tables are an important descriptive tool, they are incomplete and should be complemented by also providing:

1. A graphical representation of the frequency distribution of the development times;

2. The raw dataset, reporting the life traits of each single individual, as supplementary material.

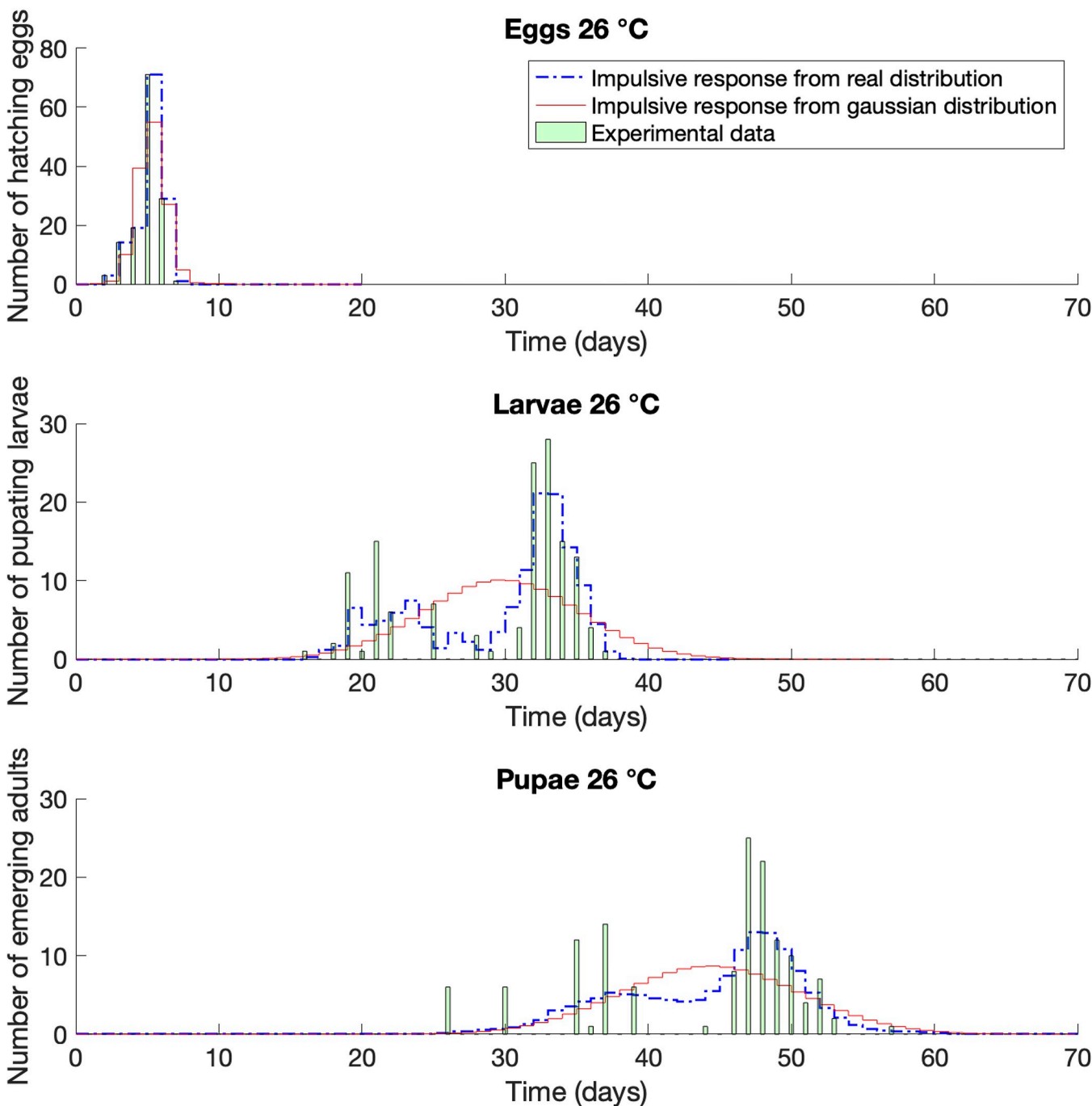

**Fig 4. Impulse response compared with life tables raw dataset [32].** Case of *Corcyra cephalonica* at constant temperature of 26˚C.

This conclusion is substantiated by the fact that the Gaussian distribution which is implied by reporting only the average and standard error in the current life tables, often does not describe reliably the actual distribution of the data. This is a quite relevant problem as it might hide interesting aspects of the ectotherms' biology (e.g., minimum development time, genetic variability, multimodal behaviour to enhance survival). In this paper we also showed that this might impact the prediction capability of models solely based on life tables parameters.

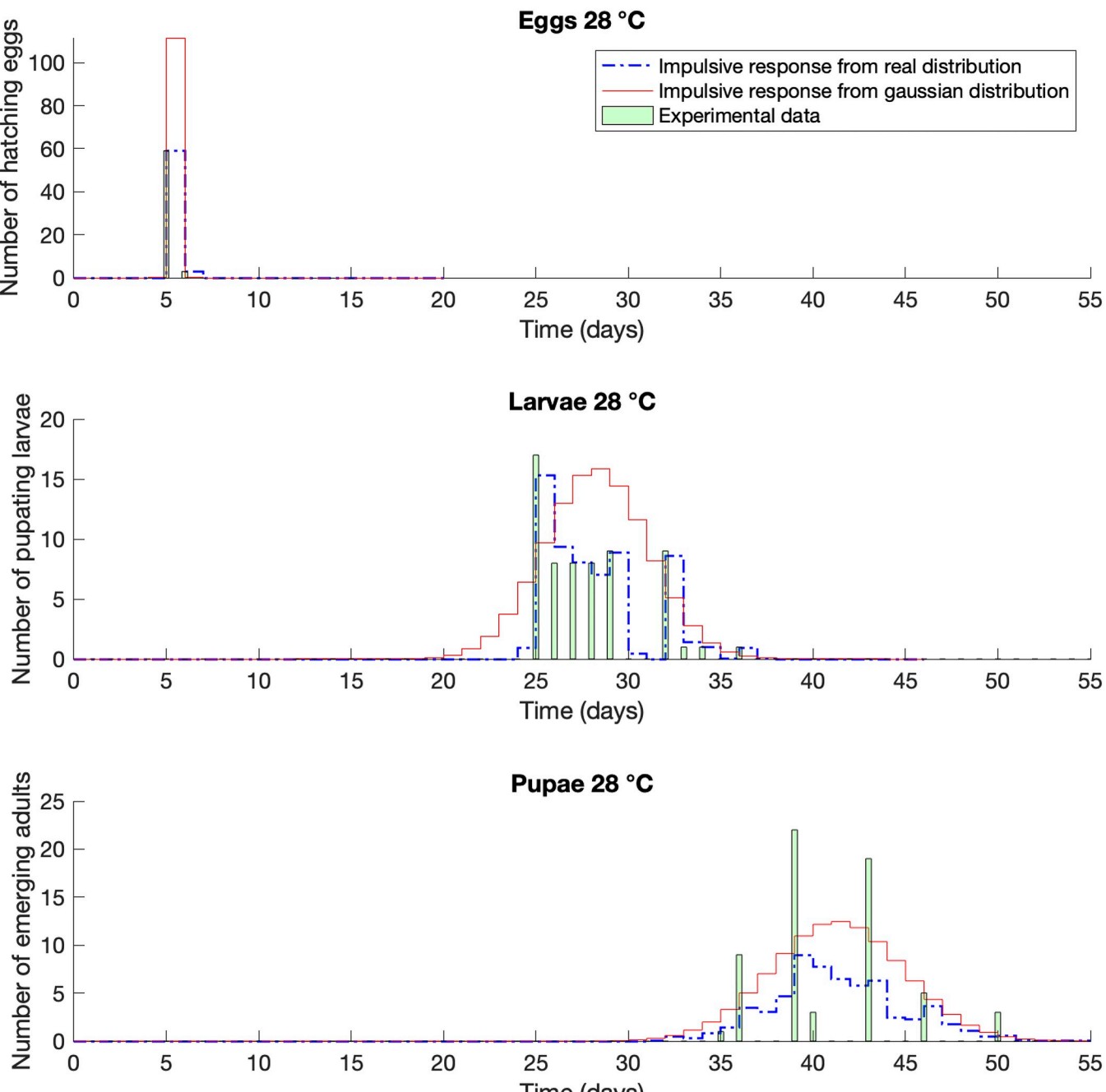

**Fig 5. Impulse response compared with life tables raw dataset [32].** Case of *Corcyra cephalonica* at constant temperature of 28˚C.

Of course, the laboratory conditions typical of life tables experiments differ significantly from natural environments, where additional factors such as agronomic practices (e.g., irrigation, pruning, harvest, fertilisation) or biotic and abiotic stresses (e.g., meteorological events, natural enemies) typically influence population development. However, the introduction and control of single factors represent the primary strength of life tables experiments providing, for instance, precise information on how plant extracts, plant metabolites, or pesticides might affect fertility, lifespan, or sex ratio [3,6].

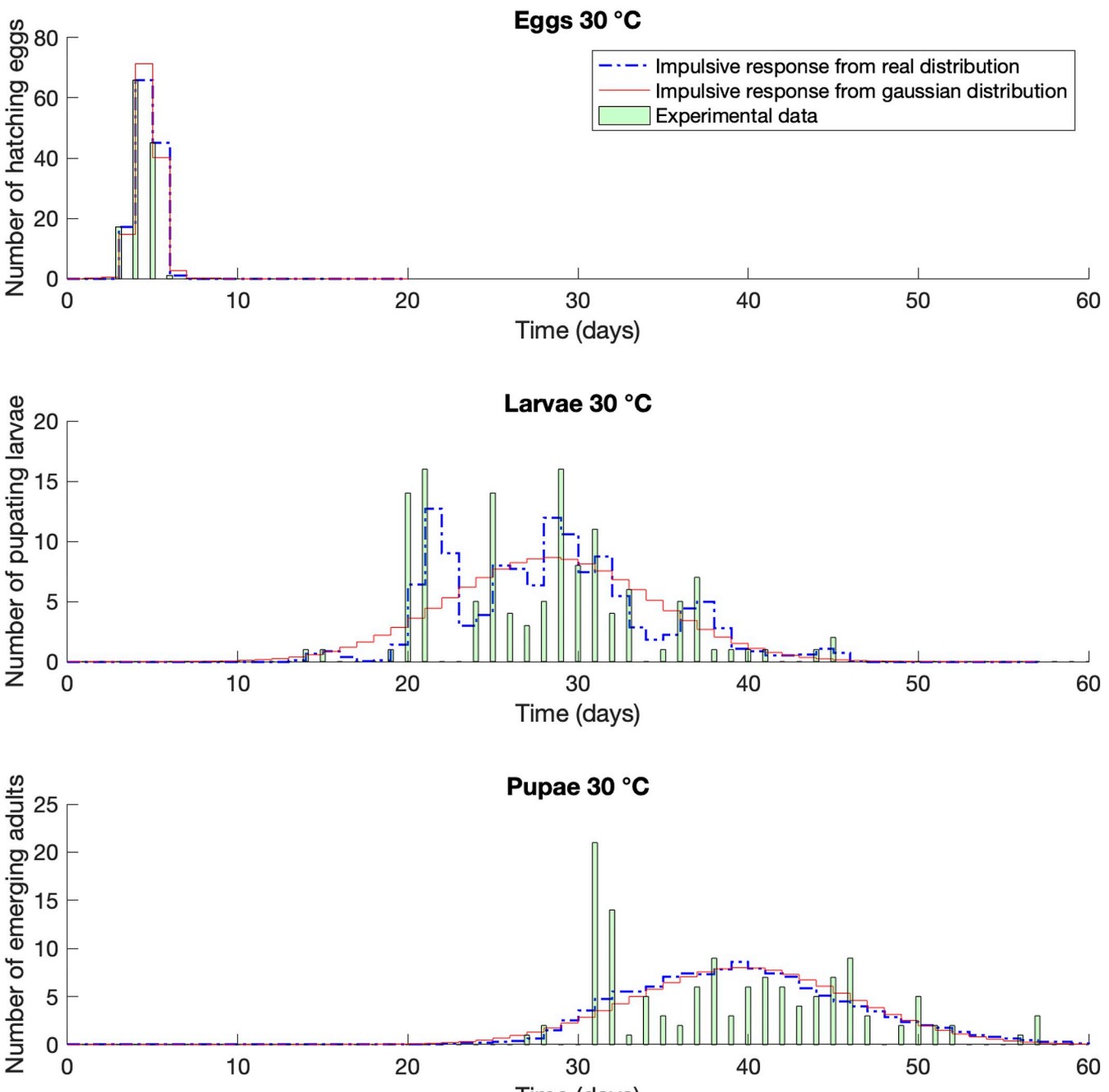

**Fig 6. Impulse response compared with life tables raw dataset [32].** Case of *Corcyra cephalonica* at constant temperature of 30˚C.

A limit of this study is that it is based on data coming from a single species, and, although it seems highly unlikely, it could be a species-specific anomaly. This however reinforces one of the main conclusions of this paper, i.e., the importance for the community to make the raw datasets publicly accessible. Indeed, publicly accessible data would not only allow the scientific community to get the actual distribution obtained for each experiment but would also allow the development of meta-analyses (e.g., refining the distributions on a single species using

experiments carried out by different laboratories, verifying similarities and differences in behaviour between organisms within the same order, genus, etc.) which at the moment are not possible.

It should also be mentioned that the experiments to complete life tables are very time consuming and require important resources, which are often provided by public agencies for the overall advancement of science and knowledge. As a consequence, making data publicly available (after a reasonable embargo period if needed) should become a standard procedure. Note that this recommendation is perfectly in line with the various Open Data policies promoted by most governmental agencies and scientific organisations.

Beside the "reporting" part, the conclusions of this paper also raise some questions concerning the practice of interpolating the mean development times reported in life tables using mathematical functions of the temperature. Indeed, the result of this paper seems to suggest that it would be much more useful and descriptive to try to interpolate the actual distributions to obtain a distribution of the development times over temperature in a 3-D surface. We believe that future explorations, possibly supported by a sufficiently large amount of publicly available data, may lead to a revision of the current interpolation framework providing a more structured theory.

## Supporting information

**S1 Fig. Quantile-quantile plot of the quantiles of the quantiles of the experimental data.** Crossed dot markers indicates the data points, while the solid reference line connects the first and third quartiles of the data and a dashed reference line extends the solid line to the ends of the data.
(PDF)

**S1 File. Dataset and script to fully reproduce the results of this study.**
(ZIP)

## Acknowledgments

The authors are grateful to the anonymous reviewers for their comments and suggestions, which have been greatly helpful for the improvement of this manuscript. The authors are grateful to Prof. Lidia Limonta and Prof. Daria Patrizia Locatelli for sharing the dataset analysed to show the core problem of this study.

## Author Contributions

**Conceptualization:** Luca Rossini, Mario Contarini, Stefano Speranza, Serhan Mermer, Vaughn Walton, Frédéric Francis, Emanuele Garone.

**Data curation:** Luca Rossini.

**Formal analysis:** Luca Rossini.

**Funding acquisition:** Mario Contarini, Stefano Speranza, Emanuele Garone.

**Investigation:** Luca Rossini.

**Methodology:** Luca Rossini, Emanuele Garone.

**Project administration:** Emanuele Garone.

**Resources:** Emanuele Garone.

**Software:** Luca Rossini.

**Supervision:** Emanuele Garone.

**Validation:** Luca Rossini.

**Visualization:** Luca Rossini.

**Writing – original draft:** Luca Rossini, Emanuele Garone.

**Writing – review & editing:** Luca Rossini, Mario Contarini, Serhan Mermer, Vaughn Walton, Frédéric Francis, Emanuele Garone.

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
