## [Decision Letter · Decision Letter 0]

12 Dec 2023

PONE-D-23-34793Life tables in entomology: a discussion on table’s parameters and the importance of raw data.PLOS ONE

Dear Dr. Rossini,

Thank you for submitting your manuscript to PLOS ONE. After careful consideration, we feel that it has merit but does not fully meet PLOS ONE’s publication criteria as it currently stands. Therefore, we invite you to submit a revised version of the manuscript that addresses the points raised during the review process.

We look forward to receiving your revised manuscript.

Kind regards,

Ramzi Mansour

Academic Editor

PLOS ONE

Journal Requirements:

"LR is funded by the European Commission under the Grant n. 101102281, Project “PestFinder”, call HORIZON-MSCA-2022-PF-01"

"The authors are grateful to Prof. Lidia Limonta and Prof. Daria Patrizia Locatelli for sharing the dataset analysed to show the core problem of this study. LR is funded by the European Commission under the Grant n. 101102281, Project “PestFinder”, call HORIZON-MSCA-2022-PF-01. Part of this work has been supported by the Fons de la Recherche Scientifique-FNRS under the Grant n. 40003443 (“Smart Testing”)."

"LR is funded by the European Commission under the Grant n. 101102281, Project “PestFinder”, call HORIZON-MSCA-2022-PF-01"

"The authors (all) declare that they have no conflict of interest."

6. We note that you have indicated that data from this study are available upon request. PLOS only allows data to be available upon request if there are legal or ethical restrictions on sharing data publicly. For more information on unacceptable data access restrictions, please see http://journals.plos.org/plosone/s/data-availability#loc-unacceptable-data-access-restrictions. 

Reviewers' comments:

Reviewer's Responses to Questions

**Comments to the Author**

1. Is the manuscript technically sound, and do the data support the conclusions?

Reviewer #1: Yes

Reviewer #2: Yes

Reviewer #3: Yes

2. Has the statistical analysis been performed appropriately and rigorously? 

Reviewer #1: Yes

Reviewer #2: N/A

Reviewer #3: Yes

3. Have the authors made all data underlying the findings in their manuscript fully available?

Reviewer #1: Yes

Reviewer #2: Yes

Reviewer #3: Yes

4. Is the manuscript presented in an intelligible fashion and written in standard English?

Reviewer #1: Yes

Reviewer #2: No

Reviewer #3: Yes

5. Review Comments to the Author

Reviewer #1: 

This paper presents a case study of the importance of raw data to the construction of life tables for modelling the development and emergence of insect pests, using a published dataset for a species of moth as a case study. The typical approach of using summary statistics (means, standard error) to model development, assuming that the data are normally distributed and therefore can be adequately represented by such values, is shown to miss important information and led to erroneous predictions. The authors therefore advocate for informing modelling efforts with raw data and that more studies in entomology to make their raw data available for future use. The points raised are valid and important, and the methods and analyses are valid, and the paper is well-written and novel. The importance of inter-individual variability, examining the actual distribution of data, and the push toward Open Data, are all in line with current trends in research. The paper will mainly be of interest to entomologists and agricultural scientists, although it will also have wider interest to developmental biologists and biostatisticians, and thus is appropriate for this journal. I believe this is an important paper that can and should be published, with some minor revisions and possible additional discussion points that I have noted below.

Title: Should be “a discussion on tables’ parameters” or “a discussion on table parameters”

Line 19: “a case of study” should be changed to “a case study”

Line 44: “Change “each temperature of rearing” to “each rearing temperature”

Line 47: Delete citation of [9] here, as it is cited below for the specific relevant points.

Lines 67-69: More importantly, the transformation from development times to rates removes the infinities that occur at developmental threshold (min/max) temperatures.

Lines 77 and 232: I think “ulteriorly” is incorrect or misleading here, it would be better to say “alternatively”.

Line 147: It might be helpful to specify here that Figure 1 shows data specifically for the larval phase at a particular temperature, as an example of the non-normal distribution of data.

Line 146: Change “accordingly” to “according”

Equations 6 and 7, and Lines 226-227, and 231: The symbols of the parameters for the Gaussian distribution and its standard error (plus the alpha symbol that should be at line 231) have disappeared from the uploaded manuscript file – please ensure that these are included and visible in the future version(s) of the paper uploaded.

Lines 241 and elsewhere: This should probably be “impulse response”, not “impulsive response”

Lines 254-256: Should restate “We remind the most interested readers to” as “We refer the most interested readers to”. Also, one wonders if maybe some of these details should be included in the main Methods section of the paper, rather than in a Supplement – not full scripts, but the packages and functions used for the tests performed may be relevant to include.

Figure 1: I was struck by the apparent bimodality of the data presented in Figure 1 – clearly this feature of the data would be lost through the use of simple Gaussian summary statistics (means, etc.) and presents a strong case for the use of raw data in this study. However, the bimodality of these data are not discussed in the paper, though multimodal behavior is mentioned briefly near the end as a general point (line 420). While somewhat beyond the main focus of the paper, one wonders how such bi/multimodality might emerge beyond just “genetic variability” – for instance, are there alternative developmental pathways or strategies (fast vs. slow growing) in the same population as a form of bet-hedging, could the eggs used have different developmental histories, etc.? Has such bi/multimodality been reported before, and how was it explained then? Perhaps the authors can draw attention to the bimodality of data in Figure 1 and provide a short (2-3 sentences) discussion of bi/multimodal development in insects in the paper?

Lines 293-303: While directly working from the actual distributions of experimental data is likely the ideal approach to capture variation and non-normality in life table parameters, as done in this study, there are some alternative (and possibly simpler) approaches that might be worth consideration. The authors mention application of alternative distributions as one alternative approach here. A related method might be data transformation – for instance, in cases where data can be made normal/Gaussian following a mathematical transformation (logarithm, square-root, etc.) it may be reasonable to use the mean and standard error values of transformed data. This approach will have limitations (for example, it likely cannot help with bi/multimodal data), but may be worth discussing. Another method was recently published using Bayesian approaches, which may be worth reading, discussing, and citing – see Studens, Bolker & Candau, 2023, “Predicting the temperature-driven development of stage-structured insect populations with a Bayesian Hierarchical Model”, Journal of Agricultural, Biological, and Environmental Statistics, https://doi.org/10.1007/s13253-023-00581-y

Line 340: There are typos in the name of this species – it should be Chrysophtharta bimaculata (Olivier, 1807) [the year may be omitted in some publications, but note the spelling of the species’ part of the name and capitalization of Olivier]

Line 344: What is “intra-genetic variability of the populations”? Maybe this should be “intra-population genetic variation”?

Lines 358-367: Often data at more extreme temperatures and/or later developmental stages can be more variability and less normally distributed due to decreased sample sizes when individuals die or fail to develop beyond earlier stages. What is interesting about this study is that the dataset used has equal sample sizes across stages (Table 1) within each temperature group and excluded extreme temperatures where nor all stages could be completed (line 128 ff.), and yet the same sorts of patterns were still found. This may speak to something intrinsic in insect development actually being captured, rather than just methodological artefacts. I suggest the authors bring these points up in the Discussion.

Reviewer #2: 

Authors provide strong arguments why the current standards around life table experiments and data may lead to important errors in models and decision support tools for pest control. Using one case study, they demonstrate that the unavailability of raw data, only publicly summarized y too coarse distribution metrics such as mean and standard errors are insufficient to accurately predict metrics of highest importance for decision support systems such as the mean generation time. This article is unambiguously of highest interest to the community of entomologists and deserves to be largely publicized. However, I have some main reservations, as well as a number of minor comments, listed below:

1/ the targeted audience is somewhat unclear, oscillating from mathematicians and modellers actually building the DSS tools and entomologists producing the primary data. While it is absolutely critical that the second group should be targeted by this paper, it somewhat misses its target at the moment, because it lacks relevant biological information: the consequences and main results should be clearly supported by illustrative examples biologically meaningful. For instance, what does it mean for an insect species if its development time distribution is bimodal versus gaussian ? Relative to the first group – DSS modellers: it is obvious that authors did not wish to push their study up to actually demonstrating the potentially catastrophic consequences of the systematic gaussian assumptions of developmental distributions on the success of pest management. However, this is a bit of a shame as it could probably have been done at minimal costs using even the most simplest DSS; one way around could be to provide more quantitative indications on how much these erroneous assumptions on distributions may crucially affect the accuracy, and therefore success of DSS tools. Put it simply: how do we care if a DSS might be 2-days wrong in predicting the peak of abundance of a pest ?

2/ how authors present the concept of distribution is somewhat limited, all mathematically (continuous), in terms of simulations (e.g., discrete individuals), and in terms of insect biology. For instance, it would be really useful and attract wider interest to relate simple characteristics of distributions to biological mechanisms. For instance, the minimum time laps between egg laying and egg hatching (ontogeny) is strongly determined genetically and biochemically due to a sequence of biochemical changes at the molecular and cellular level; injecting more biology would help entomologists understand why it matters to record and publish data in a more appropriate way.

3/ Inherent to the accuracy of the estimation of the distribution is the sampling size, which is never discussed throughout the paper, and clearly lacking. How much do the uncertainties arising from erroneous assumptions on distribution come from insufficient sampling size ? In other words, even if raw data are published, how much DSS outputs may be limited by initially insufficient sampling sizes, and therefore poor estimates around timing and other variables of key interest?

4/ the other side of the distribution/timing coin is the distribution/relative abundance aspect, never discussed either. What is more important for DSS accuracy: to predict the peak timing, or to predict the peak intensity ? The same goes to the spread of the distribution, barely touched, and which would be made more complex in a realistic scenario of overlapping generations/cohorts.

5/ The conclusion should rebound to the fact that the main conclusion most certainly hold true for other life-table metrics such as fecundity.

Detailed comments:

L20 “the Gaussian approximation of development time”

L35, L73 and elsewhere. I was surprised to see the most recent synthesis paper by Chi et al. not cited here:

Chi, H., Kavousi, A., Gharekhani, G., Atlihan, R., Özgökçe, M. S., Güncan, A., ... & Fu, J. W. (2023). Advances in theory, data analysis, and application of the age-stage, two-sex life table for demographic research, biological control, and pest management. Entomologia Generalis, 43(4), 705-35. https://doi.org/10.1127/entomologia/2023/2048

L39 “age-stage distribution” please provide a very brief definition

L47-48 “net reproduction rate” please cite (and number) the corresponding equation.

L82 What is mean by the “shape” of the distribution is unclear; in itself it may require multiple parameters to be described properly, depending on multimodality etc.

L83-85 This sentence fails its goal: of course we want to know why and how the minimal and maximal development times and shape of distribution are important for planning pest control actions, but this sentence only says that “it is important”, not “how” it is important based on a specific example. More biology is needed here !

L86-88 It is certainly not true for other related individual-based metrics such as e.g. thermal limits CTmax, and for which a bunch of studies have been / are investigating the relationship between intraspecific genetic variability and adaptation potential. It would be worth mentioning some similar examples here, emphasizing that biologists in general, and entomologists in particular, do know how to research these questions and find compelling evidence that the shape of trait distribution does matter from an ecological to an evolutionary perspective.

See e.g.,

Hoffmann AA, Chown SL, & Clusella-Trullas S (2013) Upper thermal limits in terrestrial ectotherms: how constrained are they?. Functional Ecology, 27, 934–949. https://doi.org/10.1111/j.1365-2435.2012.02036.x

but many other references are available.

L99 Mathematically and biologically, this is weird: at the individual level, there may be an experimental uncertainty on the estimate of the development time, but the mean and standard error of development times are population-level, not individual-level, metrics. This should be carefully clarified throughout.

L101-102 I read this sentence about 5 times, still can’t make sense of it. Please rewrite.

L106-113 this somewhat conclusion statement could be written more concisely: we got the point already and this will be repeated throughout.

L160 “the peak”, but also the range !!!

L169-173 This is too long and repeats information that has already been provided earlier; I suggest this should be written more concisely.

L177-178 the concept of “system identification” is unclear. Please define with a practical example. The same goes to the “impulse response identification experiment in L179.

L193 Z has not been defined...

L199 “specific datasets of C. cephalonica”

L209 How does the size of the dataset comes in there ?

L213-216 This has already been said earlier

L217-218 Add reference to figure 1 here

L235-236 kurtosis and skewness in particular can be calculated mathematically in a variety of ways, what has been used here ? It would also be good to illustrate how this has been calculated based on a schematic representation or a real data distribution.

L238-239 This statement is obvious and yet uninformative: what magnitude of deviation can be considered problematic, how far is “far” ? How should it be related to other intrinsic characteristics of data such as nominal time step and sampling size ?

L251-256 This can be written more concisely: “For the sake of completeness, the shared scripts extend the analysis to all eight temperatures.. they also include the list of all software packages and functions...”

L261-264 Already said before.

L265-268 I do not think that this introduction to a sub-subsection of a relatively short paper is necessary. I suggest cut but reintroduce the missing details in the following paragraphs.

L282 but also a shift in intensity/abundance as well…

L285 “by the pupae” please check the structure of the sentence

L290 Here it would be interesting to tell more about how these false assumptions based on data deficiency could make models outputs wrong, and how much wrong.

L292&294 and other references “Wagner et al.”

L302-303 But also accuracy on distribution shape is also constrained by sampling size no matter how many descriptive summary statistics are retained.

L305 This is good, a very clear recommendation

L310-312 The point has already been made and is clear from above, this is a repetition.

L328 “gestation lengths of cattle”

L328 “In this case, authors also plotted”

L342 “missing the life traits of the single specimens”. What ? It seems to me that the English is awkward here, doubled by a nonsensical wording.

L345 “availability of raw data”

L351 “2 days” this should be related to biological knowledge

L356 this is true only in the unrealistic case of non-overlapping generations/cohorts

L363-364 this is tautological and nonsensical, and misses to provide a mechanistic explanation; simple biological stuff such as, the higher the deviation from optimal temperature, the larger the potential damage on the individual (which is often individual-dependent, e.g. in case of large inter-individual variation in size) and therefore the larger the variability in development time due to the induction of other biochemical processes (e.g., defences/protection)

L365-366 this lacks references, this is very much a description of a bet-hedging strategy. Also lacks eco-evolutionary and biological context.

L379 “the peak of the larvae occurred” is both awkward English and lacks accuracy – the word “gaussian” should be in there.

L381 “anticipated the emergence of the first larvae”

L379-383 differences in abundances – or spread /variation across individuals within the cohort should be discussed too.

L392 How sensitive are the models and DSS to timing and abundance ? When will these erroneous assumptions and data deficiency matter or not ?

L413 “Although life tables are important…”

L421-422 you did show that this might impact the prediction capability although the “very negatively” is likely too strong since there is no quantitative statement on impacts on prediction capability/accuracy; in addition you did not show the quantitative consequences for the success of pest control; it would be good to elaborate more on this.

L423 “A limit of this study is that it is based on”

L424 “This however reinforces”

FigS1 legend “quantile-quantile plot of the quantiles of the experimental data”

In addition, I suggest using different line width and/or colours for continuous versus dashed portions, since it is currently quite difficult to visualized due to overplotting data points.

Reviewer #3: 

The work questions the completedness of information provided by common life tables in the study of Entomology, using Corcyra cephalonica as case study. The experimental question posed by the authors is of paramount importance, and the data overall support the initial hypothesis that analyzing raw data and incorporating real data distribution plots is crucial to improve the accuracy and comprehensiveness of life table analyses.

I have just minor concerns/remarks:

• I believe that the manuscript would benefit from an English language revision to improve its proficiency.

• I appreciate the fact that the authors mentioned the limitation of considering only one species (Lines 423-424). However, it would be better if the authors could add data from of at least one or two additional species. This addition would strengthen their conclusions.

• I would like to emphasize the importance of considering longevity and/or mortality, given their importance in life tables. In my opinion, the inclusion of these aspects would significantly enhance the completeness of the study.

• A critical aspect authors did not stress is that their assumptions are only true under controlled growing conditions. However, under field conditions, an insect’s development rate could be influenced by other factors, encompassing not only environmental factors, but also agronomic practices, biotic and abiotic factors, and interactions with other species.

It would be appreciated if the authors add few lines in the discussion part to stress/highlight this aspect.

Here some additional suggestions/comments:

• Abstract (Lines 16-24): In my opinion, the abstract is relatively short (149 words), falling below the 300-word limit. I would like to suggest that the authors consider expanding it by highlighting the methodology adopted and the main results of the paper, in order to provide a more robust and comprehensive version.

• Introduction: Lines 109-113 report some conclusions. I suggest removing it from the introduction section.

• Acknowledgments (Lines 444-447): According to Plos One guidelines, funding sources should not be included in the Acknowledgments.

• The references should be revised to adhere to the specified style, including the abbreviation of journal names.

6. PLOS authors have the option to publish the peer review history of their article (what does this mean?). If published, this will include your full peer review and any attached files.

Reviewer #1: **Yes: **Brady K. Quinn

Reviewer #2: No

Reviewer #3: No

---

## [Author Response · Author response to Decision Letter 0]

25 Jan 2024

Manuscript ID: PONE-D-23-34793

Title: Life tables in entomology: a discussion on tables’ parameters and the importance of raw data.

Journal: PLOS ONE

Decision letter and response to the Editor

Dear Dr. Rossini,

Thank you for submitting your manuscript to PLOS ONE. After careful consideration, we feel that it has merit but does not fully meet PLOS ONE’s publication criteria as it currently stands. Therefore, we invite you to submit a revised version of the manuscript that addresses the points raised during the review process.

We look forward to receiving your revised manuscript.

Kind regards,

Ramzi Mansour

Academic Editor

PLOS ONE

Response:

Dear Dr. Mansour,

Thank you for your time and for the possibility to reconsider our manuscript PONE-D-23-34793 for publication in PLOS ONE after a revision. We sincerely appreciated all the positive comments and suggestions provided by the Reviewers and a point-by-point response to all the questions is provided below this document. During the revision we carefully addressed all the suggestions, with the hope to have sufficiently increased the quality of the manuscript. We renew our availability for any further question or request, if needed. Thank you again for considering our manuscript.

Editor:

Journal Requirements:

Response: Thank you for this suggestion. During the revision we have formatted the document according to the style requirements of PLOS ONE, including the file naming.

Editor:

Response: Thank you for this suggestion. In the revised manuscript we have updated, according to the guidelines, the:

 Funding Information: LR is funded by the European Commission under the Grant n. 101102281, Project “PestFinder”, call HORIZON-MSCA-2022-PF-01. Part of this work has been supported by the Fons de la Recherche Scientifique-FNRS under the Grant n. 40003443 (“Smart Testing”) and by the Brussels Institute of Advanced Studies (Grant BrIAS2024).

 Financial Disclosure: The funders had no role in study design, data collection and analysis, decision to publish, or preparation of the manuscript.

These statements have been removed from the main text and provided in the cover letter, as suggested. We hope that this change fits the PLOS ONE’s guidelines.

Editor:

"LR is funded by the European Commission under the Grant n. 101102281, Project “PestFinder”, call HORIZON-MSCA-2022-PF-01"

Response: Thank you for this suggestion. As already stated in the previous answer, we have updated the Financial Disclosure and the Funding Information. These statements have been removed from the main text and included in the cover letter.

Editor:

"The authors are grateful to Prof. Lidia Limonta and Prof. Daria Patrizia Locatelli for sharing the dataset analysed to show the core problem of this study. LR is funded by the European Commission under the Grant n. 101102281, Project “PestFinder”, call HORIZON-MSCA-2022-PF-01. Part of this work has been supported by the Fons de la Recherche Scientifique-FNRS under the Grant n. 40003443 (“Smart Testing”)."

"LR is funded by the European Commission under the Grant n. 101102281, Project “PestFinder”, call HORIZON-MSCA-2022-PF-01".

Response: Thank you for this suggestion. During the revision we have modified the Acknowledgements section by removing any reference to fundings. The modification of the financial disclosure and funding information have been discussed in the previous answers.

Editor:

"The authors (all) declare that they have no conflict of interest."

Please complete your Competing Interests on the online submission form to state any Competing Interests. If you have no competing interests, please state "The authors have declared that no competing interests exist.", as detailed online in our guide for authors at http://journals.plos.org/plosone/s/submit-now.

Response: Thank you for this suggestion. During the revision we have removed the Competing Interest statement from the main text and we have provided the revised version on the cover letter.

Editor:

6. We note that you have indicated that data from this study are available upon request. PLOS only allows data to be available upon request if there are legal or ethical restrictions on sharing data publicly. For more information on unacceptable data access restrictions, please see http://journals.plos.org/plosone/s/data-availability#loc-unacceptable-data-access-restrictions. 

Response: Thank you for your suggestion. All the data to fully reproduce the results of this study, as well as all the additional data and information that may be helpful for the scientific community is publicly available at the following link https://github.com/lucaros1190/LifeTablesIssues. The dataset is provided under Creative Common Licence CC BY 4.0 DEED - Attribution 4.0 International (https://creativecommons.org/licenses/by/4.0/) and has been resubmitted to PLOS ONE as Supporting Information files for this paper. There are no restrictions on the access of the dataset. The revised statement about the data availability has been provided in the cover letter as well. A sentence in the former version of the manuscript concerning data availability was misleading and has been corrected.

Editor:

7. If there are ethical or legal restrictions on sharing a de-identified data set, please explain them in detail (e.g., data contain potentially sensitive information, data are owned by a third-party organization, etc.) and who has imposed them (e.g., an ethics committee). Please also provide contact information for a data access committee, ethics committee, or other institutional body to which data requests may be sent.

Response: According to the legislation in the Country where the experiments were carried out and the internal ethical standards of the academic institutions where the data collection was carried out. No unauthorized information on the researchers involved in the data collection is present in the data.

Editor:

Response: Thank you for this suggestion. As mentioned, above all the data are available on https://github.com/lucaros1190/LifeTablesIssues under Creative Common Licence CC BY 4.0. The data are also submitted as Supporting Information files to PLOS ONE.

Editor:

Response: Thank you for this suggestion. The revised version of the manuscript includes the Supporting Information files at the end of the manuscript, with proper in-text citations.

We hope that this revised version of the manuscript fits the PLOS ONE’s guidelines, and we renew our availability for any further change of request, if needed.

Reviewers’ comments:

Response to Reviewer 1

Reviewer 1: This paper presents a case study of the importance of raw data to the construction of life tables for modelling the development and emergence of insect pests, using a published dataset for a species of moth as a case study. The typical approach of using summary statistics (means, standard error) to model development, assuming that the data are normally distributed and therefore can be adequately represented by such values, is shown to miss important information and led to erroneous predictions. The authors therefore advocate for informing modelling efforts with raw data and that more studies in entomology to make their raw data available for future use. The points raised are valid and important, and the methods and analyses are valid, and the paper is well-written and novel. The importance of inter-individual variability, examining the actual distribution of data, and the push toward Open Data, are all in line with current trends in research. The paper will mainly be of interest to entomologists and agricultural scientists, although it will also have wider interest to developmental biologists and biostatisticians, and thus is appropriate for this journal. I believe this is an important paper that can and should be published, with some minor revisions and possible additional discussion points that I have noted below.

Response: Dear Reviewer 1, thank you very much for the time you dedicated to revise our manuscript, as well as for the helpful comments and suggestions provided with your revision. We sincerely appreciate your positive idea about our study, and we are glad to know that all the key messages are clear. During the revision process we have carefully addressed all the comments and suggestions provided, and a detailed point-by-point response follows below this message. We hope that this revised version of the manuscript fits with your expectations and we renew our availability for any further question or request, if needed. Thank you again.

Reviewer 1: Title: Should be “a discussion on tables’ parameters” or “a discussion on table parameters”

Response: Thank you for this suggestion. We have changed the title accordingly.

Reviewer 1: Line 19: “a case of study” should be changed to “a case study”

Response: Thank you for this suggestion. We have corrected this sentence accordingly.

Reviewer 1: Line 44: “Change “each temperature of rearing” to “each rearing temperature”

Response: Thank you for this suggestion. We have corrected this sentence accordingly.

Reviewer 1: Line 47: Delete citation of [9] here, as it is cited below for the specific relevant points.

Response: Thank you for this suggestion. We have updated the reference citation accordingly.

Reviewer 1: Lines 67-69: More importantly, the transformation from development times to rates removes the infinities that occur at developmental threshold (min/max) temperatures.

Response: Thank you for this comment. This information is really helpful to better understand the use of the development rates, so that the revised version of the manuscript has been integrated accordingly.

Reviewer 1: Lines 77 and 232: I think “ulteriorly” is incorrect or misleading here, it would be better to say “alternatively”.

Response: Thank you for this comment. We have replaced “ulteriorly” with “further”.

Reviewer 1: Line 147: It might be helpful to specify here that Figure 1 shows data specifically for the larval phase at a particular temperature, as an example of the non-normal distribution of data.

Response: Thank you for this suggestion. We have integrated the information of the figure caption accordingly.

Reviewer 1: Line 146: Change “accordingly” to “according”

Response: Thank you for this suggestion. There was no word “accordingly” in line 136, but probably you were referring to line 176, where effectively the suggested correction was appropriate. Thank you.

Reviewer 1: Equations 6 and 7, and Lines 226-227, and 231: The symbols of the parameters for the Gaussian distribution and its standard error (plus the alpha symbol that should be at line 231) have disappeared from the uploaded manuscript file – please ensure that these are included and visible in the future version(s) of the paper uploaded.

Response: Thank you for pointing out this issue. The problem was probably related to the submission system, because the word document seems to not have problems, while the automated PDF provided by the system does. We hope that this issue has been fixed, we will keep an eye on this issue at submission time and in case the issue reappears we will contact the support centre. 

Reviewer 1: Lines 241 and elsewhere: This should probably be “impulse response”, not “impulsive response”

Response: Thank you for pointing out this issue. We have replaced “impulsive” with “impulse” in the whole manuscript. The misprint was only in lines 241 and 369.

Reviewer 1: Lines 254-256: Should restate “We remind the most interested readers to” as “We refer the most interested readers to”. Also, one wonders if maybe some of these details should be included in the main Methods section of the paper, rather than in a Supplement – not full scripts, but the packages and functions used for the tests performed may be relevant to include.

Response: Thank you very much for this suggestion. We have addressed the suggested change and mentioned the functions/commands involved in the calculation. In our opinion, the best option is to refer directly to the code, given that the functions involved are typical of the basic environments of Matlab and R Studio. However, for the sake of completeness, we have added the main functions involved.

Reviewer 1: Figure 1: I was struck by the apparent bimodality of the data presented in Figure 1 – clearly this feature of the data would be lost through the use of simple Gaussian summary statistics (means, etc.) and presents a strong case for the use of raw data in this study. However, the bimodality of these data are not discussed in the paper, though multimodal behavior is mentioned briefly near the end as a general point (line 420). While somewhat beyond the main focus of the paper, one wonders how such bi/multimodality might emerge beyond just “genetic variability” – for instance, are there alternative developmental pathways or strategies (fast vs. slow growing) in the same population as a form of bet-hedging, could the eggs used have different developmental histories, etc.? Has such bi/multimodality been reported before, and how was it explained then? Perhaps the authors can draw attention to the bimodality of data in Figure 1 and provide a short (2-3 sentences) discussion of bi/multimodal development in insects in the paper?

Response: Thank you for this suggestion and very insightful comment. Bimodality/Multimodality is a fascinating phenomenon that has sparked our curiosity too when we analysed the data. We have added a few sentences in the introduction and results/discussion of this revised version of the manuscript to highlight the very relevant points introduced in your comment, in particular as these phenomena reinforce our argument. We also report in the manuscript that as a matter-of-fact multi-modality has been already observed for other species, which is actually an important argument to mitigate the “one-species limitation” of our study. Concerning the observation about the “causes”, we agree with the reviewer that “continuous” genetical variability is not sufficient to explain multi-modality (which is probably also only one of the phenomena involved even in simpler distributions), but that multimodality is probably due to “swich” effects that determine different developmental pathways. Unfortunately, the current empirical evidence does not allow us to go beyond some speculation, but it might be a starting point for future studies. Thank you again for this comment which allowed us to make an even stronger argument about the need to publish raw data to keep the completeness of the information.

Reviewer 1: Lines 293-303: While directly working from the actual distributions of experimental data is likely the ideal approach to capture variation and non-normality in life table parameters, as done in this study, there are some alternative (and possibly simpler) approaches that might be worth consideration. The authors mention application of alternative distributions as one alternative approach here. A related method might be data transformation – for instance, in cases where data can be made normal/Gaussian following a mathematical transformation (logarithm, square-root, etc.) it may be reasonable to use the mean and standard error values of transformed data. This approach will have limitations (for example, it likely cannot help with bi/multimodal data) but may be worthy of discussion. Another method was recently published using Bayesian approaches, which may be worth reading, discussing, and citing – see Studens, Bolker & Candau, 2023, “Predicting the temperature-driven development of stage-structured insect populations with a Bayesian Hierarchical Model”, Journal of Agricultural, Biological, and Environmental Statistics, https://doi.org/10.1007/s13253-023-00581-y

Response: Thank you for this consideration and for the interesting reference that have been added and discussed in the manuscript. As an “offline” comment, we agree that data transformation and Bayesian approaches can be a valuable tool to analyse the data and try to extract useful information, but in many cases (at least from our personal experience) it is very difficult to find a mathematical expression that properly transforms the dataset. Moreover, it is difficult to associate an actual meaning to the transformation, which in the end often makes the approach more a “data compression tool” rather than an analysis tool.

Reviewer 1: Line 340: There are typos in the name of this species – it should be Chrysophtharta bimaculata (Olivier, 1807) [the year may be omitted in some publications, but note the spelling of the species’ part of the name and capitalization of Olivier]

Response: Thank you for pointing out this issue. We have corrected the name accordingly.

Reviewer 1: Line 344: What is “intra-genetic variability of the populations”? Maybe this should be “intra-population genetic variation”?

Response: Thank you for spotting this. Yes, the meaning of this sentence is exactly what you mean. To avoid any potential misunderstanding, we have corrected this part of the text accordingly.

Reviewer 1: Lines 358-367: Often data at more extreme temperatures and/or later developmental stages can be more variability and less normally distributed due to decreased sample sizes when individuals die or fail to develop beyond earlier stages. What is interesting about this study is that the dataset used has equal sample sizes across stages (Table 1) within each temperature group and excluded extreme temperatures where nor all stages could be completed (line 128 ff.), and yet the same sorts of patterns were still found. This may speak to something intrinsic in insect development actually being captured, rather than just methodological artefacts. I suggest the authors bring these points up in the Discussion.

Response: Thank you for this consideration. We have added in the discussion a comment highlighting that with this specific dataset extreme temperature are sufficiently covered to be reasonably confident that the non-gaussian distribution are not artifacts due to a low sample size.

Response to Reviewer 2

Reviewer 2: Authors provide strong arguments why the current standards around life table experiments and data may lead to important errors in models and decision support tools for pest control. Using one case study, they demonstrate that the unavailability of raw data, only publicly summarized y too coarse distribution metrics such as mean and standard errors are insufficient to accurately predict metrics of highest importance for decision support systems such as the mean generation time. This article is unambiguously of highest interest to the community of entomologists and deserves to be largely publicized. However, I have some main reservations, as well as a number of minor comments, listed below.

Response: Dear Reviewer 2, thank you for the time dedicated to revise our manuscript, as well as for the helpful comments and suggestions provided with the revision. We sincerely appreciate your positive opinion, and a point-by-point answer to all your questions and issues follows below. We hope that this revised version of the manuscript better reflects your expectations, and we renew our availability for any further question or request, if needed.

Reviewer 2: 1) the targeted audience is somewhat unclear, oscillating from mathematicians and modellers actually building the DSS tools and entomologists producing the primary data. While it is absolutely critical that the second group should be targeted by this paper, it somewhat misses its target at the moment, because it lacks relevant biological information: the consequences and main results should be clearly supported by illustrative examples biologically meaningful. For instance, what does it mean for an insect species if its development time distribution is bimodal versus gaussian? Relative to the first group – DSS modellers: it is obvious that authors did not wish to push their study up to actually demonstrating the potentially catastrophic consequences of the systematic gaussian assumptions of developmental distributions on the success of pest management. However, this is a bit of a shame as it could probably have been done at minimal costs using even the most simplest DSS; one way around could be to provide more quantitative indications on how much these erroneous assumptions on distributions may crucially affect the accuracy, and therefore success of DSS tools. Put it simply: how do we care if a DSS might be 2-days wrong in predicting the peak of abundance of a pest?

Response: 

Thank you for your comment. Reading again the paper we see your point. Indeed, the main message of this paper (i.e. the need for publishing raw data and the possible deceptive information given by relying only on mean and standard error) is meant for both publics: on the one hand this paper targets people interested in building models (e.g. for DSSs) to warn about the dangers of using gaussian interpretation of the data which might lead to models that are not predictive; on the other hand it wants to point out at possible missed analysis opportunities for “descriptive” entomologists as information like min and max development rate, possible unimodal/bimodal/multimodal distribution etc might give biological insight on certain species. In the new version of the manuscript these two aspects have been stressed from the very beginning of the paper.

Concerning the specific dangers for DSS modelling, following the reviewers’ suggestion we have better commented Figure 1 which clearly shows the difference between a model built using the gaussian hypothesis and the model built using the actual distribution. In this figure we comment as the peak time of a model based on the gaussian assumption had no practical meaning and could lead to wrong conclusions.

Reviewer 2: 2) how authors present the concept of distribution is somewhat limited, all mathematically (continuous), in terms of simulations (e.g., discrete individuals), and in terms of insect biology. For instance, it would be really useful and attract wider interest to relate simple characteristics of distributions to biological mechanisms. For instance, the minimum time laps between egg laying and egg hatching (ontogeny) is strongly determined genetically and biochemically due to a sequence of biochemical changes at the molecular and cellular level; injecting more biology would help entomologists understand why it matters to record and publish data in a more appropriate way.

Response: Thank you for this comment. During the revision we have warmly welcomed the suggestion of injecting more biology to provide a more practical explanation to the concerns highlighted by our study. There are different integrations throughout the text, with the hope to have sufficiently increased the biological content of the manuscript. 

Reviewer 2: 3) Inherent to the accuracy of the estimation of the distribution is the sampling size, which is never discussed throughout the paper, and clearly lacking. How much do the uncertainties arising from erroneous assumptions on distribution come from insufficient sampling size? In other words, even if raw data are published, how much DSS outputs may be limited by initially insufficient sampling sizes, and therefore poor estimates around timing and other variables of key interest?

Response: Thank you for this very relevant comment. This is a major open question concerning the experiment design of life tables. Somehow, answering this question is one of our long-term ambitions and this study is placed one step before tackling this question. Indeed, if by sufficient experience we might identify the shapes of the most typical distributions for insect development, we could plan the sample size of the experiments accordingly. However, at the moment not sufficient information is available in the literature to tackle this issue in a serious way. We believe that data sharing is fundamental to this goal: the authors have first-hand experience on the human and economic costs to rear large numbers of individuals, and getting enough data on enough different species to identify the shapes of the most common distribution is clearly unfeasible for a single research group (even for the largest ones). We strongly believe that open data is the solution, and of course, once the shape of the distribution is known it is fundamental to define the number of eggs to rear by considering the mortality and the classes of the distribution itself. We hope to have sufficiently answered your question, and that it fits your expectations. Part of these long-term ambitions are now discussed in the revised version of the manuscript.

Reviewer 2: 4) the other side of the distribution/timing coin is the distribution/relative abundance aspect, never discussed either. What is more important for DSS accuracy: to predict the peak timing, or to predict the peak intensity? The same goes to the spread of the distribution, barely touched, and which would be made more complex in a realistic scenario of overlapping generations/cohorts.

Response: Thank you for this comment. The core of this study is the distribution of the development times, not the abundance. We decided to not mention this aspect to not generate confusion: even if they are strictly related from a pest control perspective, they are independent problems to treat. Of course, the timing (i.e., the peak of the population) is important to set up a control strategy, but the abundance defines if the control action will be carried out. The greatest part of the thresholds is defined as “X number of individuals per plant”, “X number of individuals per hectare” (or similar definitions), so of course the abundance is important. From a modelling perspective, the abundance, above all in open field conditions, is of difficult estimation for different reasons, as for example migration of individuals, uncontrolled effects, and/or approximations in the model formulations. We were among the firsts to face this problem and the details and motivations are widely explained in https://doi.org/10.1016/j.ifacol.2022.11.128 and https://doi.org/10.1016/j.ecoinf.2023.102310. For the sake of completeness, we have mentioned the problem in the manuscript, but without going into the details. We hope that these changes fit your expectations.

Reviewer 2: 5) The conclusion should rebound to the fact that the main conclusion most certainly hold true for other life-table metrics such as fecundity.

Response: Thank you for this comment which allowed to extend the scope of this paper. This aspect is now mentioned in the conclusion as suggested by the reviewer.

Detailed comments:

Reviewer 2: L20 “the Gaussian approximation of development time”

Response: Thank you for this suggestion. We have corrected this part of the text accordingly.

Reviewer 2: L35, L73 and elsewhere. I was surprised to see the most recent synthesis paper by Chi et al. not cited here:

Chi, H., Kavousi, A., Gharekhani, G., Atlihan, R., Özgökçe, M. S., Güncan, A., ... & Fu, J. W. (2023). Advances in theory, data analysis, and application of the age-stage, two-sex life table for demographic research, biological control, and pest management. Entomologia Generalis, 43(4), 705-35. https://doi.org/10.1127/entomologia/2023/2048

Response: Thank you for this suggestion. At the time the first final version of the manuscript (before the first submission) was prepared, we were not aware of this paper. We have cited this paper in this revised version of the manuscript.

Reviewer 2: L39 “age-stage distribution” please provide a very brief definition

Response: Thank you for this suggestion. We have added a brief definition after the first appearance of “age-stage distribution”.

Reviewer 2: L47-48 “net reproduction rate” please cite (and number) the corresponding equation. 

Response: Thank you for this suggestion. We have numbered the equation.

Reviewer 2: L82 What is mean by the “shape” of the distribution is unclear; in itself it may require multiple parameters to be described properly, depending on multimodality etc.

Response: Thank you for this comment. This part of the manuscript has been integrated with additional information that would better introduce all the general overview. We hope that the changes fit your expectations.

Reviewer 2: L83-85 This sentence fails its goal: of course we want to know why and how the minimal and maximal development times and shape of distribution are important for planning pest control actions, but this sentence only says that “it is important”, not “how” it is important based on a specific example. More biology is needed here !

Response: Thank you for this suggestion. We have revised this part of the text by “injecting” more biology to justify the concepts discussed. We hope that our corrections fit your expectations.

Reviewer 2: L86-88 It is certainly not true for other related individual-based metrics such as e.g. thermal limits CTmax, and for which a bunch of studies have been / are investigating the relationship between intraspecific genetic variability and adaptation potential. It would be worth mentioning some similar examples here, emphasizing that biologists in general, and entomologists in particular, do know how to research these questions and find compelling evidence that the shape of trait distribution does matter from an ecological to an evolutionary perspective.

See e.g.,

Hoffmann AA, Chown SL, & Clusella-Trullas S (2013) Upper thermal limits in terrestrial ectotherms: how constrained are they?. Functional Ecology, 27, 934–949. https://doi.org/10.1111/j.1365-2435.2012.02036.x

but many other references are available.

Response: Thank you for this comment. We have modified this part of the manuscript accordingly, providing the integrations and changes suggested. We hope that this revised part of the text better suits your expectations.

Reviewer 2: L99 Mathematically and biologically, this is weird: at the individual level, there may be an experimental uncertainty on the estimate of the development time, but the mean and standard error of development times are population-level, not individual-level, metrics. This should be carefully clarified throughout.

Response: Thank you for this comment. There was an error in the sentence: the word “individual” is actually “life stage”. We changed this sentence accordingly providing its original meaning.

Reviewer 2: L101-102 I read this sentence about 5 times, still can’t make sense of it. Please rewrite.

Response: Thank you for this comment. We have modified the sentence accordingly to make it simpler and more comprehensible. We hope that this correction fits with your expectations.

Reviewer 2: L106-113 this somewhat conclusion statement could be written more concisely: we got the point already and this will be repeated throughout.

Response: Thank you for this suggestion. We have shortened this part of the text, following the suggestion of the other reviewers as well. We hope that this revised part of the text better suits your expectations.

Reviewer 2: L160 “the peak”, but also the range !!!

Response: Thank you for this comment. We have modified this part of the text accordingly. We hope that now it is clearer.

Reviewer 2: L169-173 This is too long and repeats information that has already been provided earlier; I suggest this should be written more concisely.

Response: Thank you for this suggestion. We have modified this part of the text accordingly to synthesise the above-mentioned rows.

Reviewer 2: L177-178 the concept of “system identification” is unclear. Please define with a practical example. The same goes to the “impulse response identification experiment in L179.

Response: Thank you for this comment. During the revision we added a brief definition of “system identification”. We hope that this change fits with your expectations.

Reviewer 2: L193 Z has not been defined...

Response: Thank you for this comment. We have modified this part of the text accordingly rephrasing as “Z -transform domain” to avoid confusion.

Reviewer 2: L199 “specific datasets of C. cephalonica”

Response: Thank you for this suggestion. We have modified this part of the text accordingly.

Reviewer 2: L209 How does the size of the dataset comes in there ?

Response: Thank you for this comment. Being a model, the impulse response is normalised and thus independent on the sample size. In line of principle, if one wants to add in the model a measure of the uncertainty (which is related to the size of the sample) the model should be complemented by a “disturbance term” whose role is to take into account of the uncertainty (which comes from the size of the sample). However, we felt that adding such an extra model (whose parameters at the current stage cannot be estimated) would decrease dramatically the understandability of the paper distracting from the main message.

Reviewer 2: L213-216 This has already been said earlier

Response: Thank you for this comment. We have removed the above-mentioned lines.

Reviewer 2: L217-218 Add reference to figure 1 here

Response: Thank you for this comment. We have introduced the reference to Fig. 1.

Reviewer 2: L235-236 kurtosis and skewness in particular can be calculated mathematically in a variety of ways, what has been used here ? It would also be good to illustrate how this has been calculated based on a schematic representation or a real data distribution.

Response: Thank you for this comment. The calculation has been carried out according to the definitions of skewness and kurtosis reported in https://doi.org/10.1016/B978-190399655-3/50011-6, already included in the “kurtosis()” and “skewness()” Matlab functions. We have added the explicit formulae and the citation into the text. In our opinion, a schematic representation of the calculation is redundant, since there is the full dataset and code publicly available to the readers that want more information on this aspect. We hope that this change fits with your expectations.

Reviewer 2: L238-239 This statement is obvious and yet uninformative: what magnitude of deviation can be considered problematic, how far is “far”? How should it be related to other intrinsic characteristics of data such as nominal time step and sampling size ?

Response: Thank you for this comment. We have integrated this part of the text with a brief description of skewness and kurtosis w.r.t. some limit values. We hope that this part of the text has been sufficiently improved.

Reviewer 2: L251-256 This can be written more concisely: “For the sake of completeness, the shared scripts extend the analysis to all eight temperatures.. they also include the list of all software packages and functions...”

Response: Thank you for this suggestion. During the revision we have compressed this part of the text. We hope to have sufficiently increased the readability of this subsection.

Reviewer 2: L261-264 Already said before.

Response: Thank you for this comment. We have removed the aforementioned lines.

Reviewer 2: L265-268 I do not think that this introduction to a sub-subsection of a relatively short paper is necessary. I suggest cut but reintroduce the missing details in the following paragraphs.

Response: Thank you for this suggestion. Following the previous comment, we have shortened all the aforementioned lines, maintaining the information that in our opinion should be highlighted.

Reviewer 2: L282 but also a shift in intensity/abundance as well…

Response: Thank you for this comment. As we have explained in previous comments, talking about abundance is misleading in this case. It is obvious that if the number of total individuals is the same (normalisation factor of the distribution) and the distributions are different, one may look “higher” than the other. In our opinion there is no need to refer to the abundance, simply because the normalisation factor of the distributions is the same (Gaussian vs actual), but the shape makes the difference.

Reviewer 2: L285 “by the pupae” please check the structure of the sentence

Response: Thank you for this comment. We have corrected this sentence accordingly.

Reviewer 2: L290 Here it would be interesting to tell more about how these false assumptions based on data deficiency could make models outputs wrong, and how much wrong.

Response: Thank you for this comment. Actually, your request is the whole Section 3.2! The structure of the manuscript divides the part of the life tables experiments/analysis and of the implication of the life tables value on modelling purposes. In this line we are still talking about the data analysis, the modelling part comes after.

Reviewer 2: L292&294 and other references “Wagner et al.”

Response: Thank you for this suggestion. We corrected the citations accordingly.

Reviewer 2: L302-303 But also accuracy on distribution shape is also constrained by sampling size no matter how many descriptive summary statistics are retained.

Response: Thank you for this comment. We have modified this sentence in order to include your consideration.

Reviewer 2: L305 This is good, a very clear recommendation

Response: Thank you for this very positive comment. Actually, this is one of our main recommendations, besides sharing the raw data. Accessing the source of information is fundamental for double checking and reproducing the results of the experiments, as we have repeatedly stated within the manuscript.

Reviewer 2: L310-312 The point has already been made and is clear from above, this is a repetition.

Response: Thank you for this comment. We have removed the mentioned sentence.

Reviewer 2: L328 “gestation lengths of cattle”

Response: Thank you for this suggestion. We have corrected this sentence accordingly.

Reviewer 2: L328 “In this case, authors also plotted”

Response: Thank you for this suggestion. We have corrected this sentence accordingly.

Reviewer 2: L342 “missing the life traits of the single specimens”. What ? It seems to me that the English is awkward here, doubled by a nonsensical wording.

Response: Thank you for this comment. We have modified this sentence accordingly, with the hope to have improved its readability and meaning.

Reviewer 2: L345 “availability of raw data”

Response: Thank you for this comment. We have corrected this sentence accordingly.

Reviewer 2: L351 “2 days” this should be related to biological knowledge

Response: Thank you for this comment. Yes, it is the biological information that one can extract from the raw data or from a plot of the actual distribution.

Reviewer 2: L356 this is true only in the unrealistic case of non-overlapping generations/cohorts

Response: Thank you for this comment. Yes, of course there are more complicated cases such as overlapping generations, but in this sentence, we wanted to address the key message in a very simple way.

Reviewer 2: L363-364 this is tautological and nonsensical, and misses to provide a mechanistic explanation; simple biological stuff such as, the higher the deviation from optimal temperature, the larger the potential damage on the individual (which is often individual-dependent, e.g. in case of large inter-individual variation in size) and therefore the larger the variability in development time due to the induction of other biochemical processes (e.g., defences/protection).

Response: Thank you for this comment. We rephrased this sentence to make it clearer.

Reviewer 2: L365-366 this lacks references, this is very much a description of a bet-hedging strategy. Also lacks eco-evolutionary and biological context.

Response: Thank you for this comment. We have added references to support the statement.

Reviewer 2: L379 “the peak of the larvae occurred” is both awkward English and lacks accuracy – the word “gaussian” should be in there.

Response: Thank you for this comment. We have corrected this sentence accordingly.

Reviewer 2: L381 “anticipated the emergence of the first larvae”

Response: Thank you for this suggestion. We have corrected this sentence accordingly.

Reviewer 2: L379-383 differences in abundances – or spread /variation across individuals within the cohort should be discussed too.

Response: Thank you for this comment. As motivated before, talking about abundance is misleading for the readers and potentially out of the scope of this manuscript.

Reviewer 2: L392 How sensitive are the models and DSS to timing and abundance? When will these erroneous assumptions and data deficiency matter or not?

Response: Thank you for this question. As we showed in the simulations (Fig. 3 and section 3.2) erroneous assumptions and data deficiency matter. The abundance is a relatively solvable problem, as we explained in previous comments. A wrong estimation of the development time, instead, can be a problem because it can cause a compression/dilatation of the mean generation times. For instance, if we use the models to simulate three, four, etc.. generations and the mean development time is underestimated or overestimated, we are accumulating an anticipation or a delay in our simulation.

Reviewer 2: L413 “Although life tables are important…”

Response: Thank you for this suggestion. We have corrected this sentence accordingly. 

Reviewer 2: L421-422 you did show that this might impact the prediction capability although the “very negatively” is likely too strong since there is no quantitative statement on impacts on prediction capability/accuracy; in addition you did not show the quantitative consequences for the success of pest control; it would be good to elaborate more on this.

Response: Thank you for this comment. We have modified this sentence to make the statements softer.

Reviewer 2: L423 “A limit of this study is that it is based on”

Response: Thank you for this suggestion. We have modified this sentence accordingly.

Reviewer 2: L424 “This however reinforces”

Response: Thank you for this suggestion. We have corrected this sentence accordingly.

Reviewer 2: FigS1 legend “quantile-quantile plot of the quantiles of the experimental data”. In addition, I suggest using different line width and/or colours for continuous versus dashed portions, since it is currently quite difficult to visualized due to overplotting data points.

Response: Thank you for this suggestion. We have corrected the legend and the figure accordingly.

Response to Reviewer 3

Reviewer 3: The work questions the completedness of information provided by common life tables in the study of Entomology, using Corcyra cephalonica as case study. The experimental question posed by the authors is of paramount importance, and the data overall support the initial hypothesis that analyzing raw data and incorporating real data distribution plots is crucial to improve the accuracy and comprehensiveness of life table analyses.

Response: Dear Reviewer 3, thank you for the time dedicated to revise our manuscript, as well as for the helpful comments and suggestions provided with the revision. We have sincerely appreciated your positive comments and we have carefully addressed all the comments. A point-by-point response to all the comments follows below in this letter. We hope that the revision has sufficiently increased the quality of the manuscript, and we renew our availability for any further question or request, if needed.

I have just minor concerns/remarks:

Reviewer 3: I believe that the manuscript would benefit from an English language revision to improve its proficiency.

Response: Thank you very much for this comment. During the revision we have paid particular attention to correct the English language and style, with the hope that now the manuscript is more readable.

Reviewer 3: I appreciate the fact that the authors mentioned the limitation of considering only one species (Lines 423-424). However, it would be better if the authors could add data from of at least one or two additional species. This addition would strengthen their conclusions.

Response: Thank you for this remark. We totally agree with this comment, and we have also pointed this issue in the conclusion. However, finding row data from a different species that allows us to carry out exactly the same analysis is difficult and goes slightly out of the scope of this study. However, we extended the discussion and the insights that our dataset provided from a biological point of view as, for instance, the bimodality of larvae at 26 °C. We hope that in future we can delve into the different open questions highlighted in this study by referring to different case studies.

Reviewer 3: I would like to emphasize the importance of considering longevity and/or mortality, given their importance in life tables. In my opinion, the inclusion of these aspects would significantly enhance the completeness of the study.

Response: Thank you for this comment. Actually, the inclusion of mortality and fertility was in our initial planning, however we had to adapt the study on the data we had, which only include the overall mortality of the stages but not their temporal distribution. From the same methodology we applied for the development, it is possible to describe mortality, but it should be demonstrated with proper experimental data. Accordingly, this aspect is among the open questions that we left with this manuscript, and we hope to provide an answer soon. Following this comment, however, we have modified the impulse response model to account for the stage mortality (eqs. 6, 7).

Reviewer 3: A critical aspect authors did not stress is that their assumptions are only true under controlled growing conditions. However, under field conditions, an insect’s development rate could be influenced by other factors, encompassing not only environmental factors, but also agronomic practices, biotic and abiotic factors, and interactions with other species.

It would be appreciated if the authors add few lines in the discussion part to stress/highlight this aspect.

Response: Thank you for this comment. Of course, this is a relevant aspect of pest populations and poikilothermic individuals in general. The anthology of this work, however, it is based on discussions we had looking at laboratory data. We realized, in fact, that most of the models describing pest populations fails in representing laboratory datasets, where the condition is supposed to be optimal and less affected by uncontrolled factors. The paradox, is that these models have been successfully validated with open field data, arising the question of “where is the problem”. After some attempts, we realized that this discordance between model simulations and laboratory data is given by a compression of the generations, striking the spark from which this study was born. We think that a step back is necessary to provide a better mathematical description (maximising the amount of biological information) of insect populations. This step back is to critically re-think how to collect and analyse the experimental data from life tables studies, basic operation to investigate the essential biological traits of the species.

We have added a few lines to remark that the growth under laboratory conditions is an extremely controlled environment that may be far from the field, where more natural conditions occur. We decided to avoid mentioning the open field conditions in the first draft of the paper to not generate misunderstanding to potential readers, but during the revision we have warmly welcomed your suggestion. We hope that the integration fits with your expectations.

Here some additional suggestions/comments:

Reviewer 3: Abstract (Lines 16-24): In my opinion, the abstract is relatively short (149 words), falling below the 300-word limit. I would like to suggest that the authors consider expanding it by highlighting the methodology adopted and the main results of the paper, in order to provide a more robust and comprehensive version.

Response: Thank you for this suggestion. During the revision we have expanded the abstract with more information about the methodology and the conclusion. We hope that this revised part of the text fits your expectations.

Reviewer 3: Introduction: Lines 109-113 report some conclusions. I suggest removing it from the introduction section.

Response: Thank you for this suggestion. During the revision we have modified this part of the manuscript accordingly. More specifically, we moved part of the above mentioned text in the final part of the abstract, given that an extension was requested.

Reviewer 3: Acknowledgments (Lines 444-447): According to Plos One guidelines, funding sources should not be included in the Acknowledgments.

Response: Thank you for this suggestion. We have carefully revised all the aspects related to the journal settings, including the acknowledgements and fundings.

Reviewer 3: The references should be revised to adhere to the specified style, including the abbreviation of journal names.

Response: Thank you for this comment. As explained in the previous comment, we have carefully revised all the text settings, according to the journal guidelines.

---

## [Decision Letter · Decision Letter 1]

9 Feb 2024

PONE-D-23-34793R1Life tables in entomology: a discussion on tables' parameters and the importance of raw data.PLOS ONE

Dear Dr. Rossini,

Thank you for submitting your manuscript to PLOS ONE. After careful consideration, we feel that it has merit but does not fully meet PLOS ONE’s publication criteria as it currently stands. Therefore, we invite you to submit a revised version of the manuscript that addresses the points raised during the review process.

We look forward to receiving your revised manuscript.

Kind regards,

Ramzi Mansour

Academic Editor

PLOS ONE

Journal Requirements:

Reviewers' comments:

Reviewer's Responses to Questions

**Comments to the Author**

1. If the authors have adequately addressed your comments raised in a previous round of review and you feel that this manuscript is now acceptable for publication, you may indicate that here to bypass the “Comments to the Author” section, enter your conflict of interest statement in the “Confidential to Editor” section, and submit your "Accept" recommendation.

Reviewer #1: All comments have been addressed

Reviewer #2: (No Response)

2. Is the manuscript technically sound, and do the data support the conclusions?

Reviewer #1: Yes

Reviewer #2: Yes

3. Has the statistical analysis been performed appropriately and rigorously? 

Reviewer #1: Yes

Reviewer #2: Yes

4. Have the authors made all data underlying the findings in their manuscript fully available?

Reviewer #1: Yes

Reviewer #2: Yes

5. Is the manuscript presented in an intelligible fashion and written in standard English?

Reviewer #1: Yes

Reviewer #2: Yes

6. Review Comments to the Author

Reviewer #1: 

The authors have adequately addressed my comments and (in my opinion) those of the other reviewers, and the paper can now be accepted in its revised form. I commend them on putting together a great piece of work - I'm sure this will be an important paper in the field.

Reviewer #2: 

Authors made a great effort to address all reviewers’ comment and incorporate changes in their article, which I believe are a significant improvement. I only have very minor comments below.

L25 “the benefits”

L30 “biological aspects”

L31 avoid using “evolution” in a demographic sense. I recommend “changes in the population”, or even “demographic changes”.

L31 “highlights this by” or “highlights this aspect by” sounded better

L46 “the individuals in a population” or “from a population”

L49 why did you remove “a” before cohort ? Are you sure this is correct English in current form?

L54 “to obtain various information”

L56 “The age-stage distribution describes… that compose the insect’s life cycle.”

L76 “outside such thermal range”

L123 “this is not yet common practice, which produces a loss”

L124 “Remarkably, the unavailability of data”. A more offline comment here: a lot of papers publish a statement similar to “data are available from authors upon request”, notably because publishing data still remains a quite tedious, non straightforward and sometimes slow process, but most importantly also costly, and the community lacks general guidelines of the multiple ways to publish data. First, while this requires lots of efforts to collect data a posteriori from authors, I wonder how often authors do respond to these post-publication queries, is it something that the authors of the present paper have a sense of ? Second, wouldn’t it also be the role of publishers and journals to build up tools and guidelines about the so many available ways to publish data beyond the gold standards (e.g., Dryad), how are all the alternative (free github repositories, as supplementary materials of a publication, etc.) valid

L135 does it really assume “identical”, or instead “interchangeable” as random draws of a single population ? A random draw does not assume identity, but representativeness. The distribution will be imperfect (unrealistic) in case the draw is too small relative to the size of the population, and in case the population itself is not representative of the species. I’m not very sure I agree with the following argument L135-137; bootstrapping will smooth-out the distribution and a distribution is always based on individuals’ traits, this last sentence as currently written seems to oppose bootstrapping to individual-based data, so I don’t understand…

L147 “of insect population models”, I assume not all modellers deal with pests.

L148 “exclusive usage of life tables’ synthetic information”

L200 is it instead “implies that individuals from a single population can have two different developmental times” ?

L316 “are both publicly”

L369 “Furthermore, any synthetic parameters from descriptive statistics will always come with some loss of information”

7. PLOS authors have the option to publish the peer review history of their article (what does this mean?). If published, this will include your full peer review and any attached files.

Reviewer #1: **Yes: **Brady K. Quinn

Reviewer #2: No

---

## [Author Response · Author response to Decision Letter 1]

11 Feb 2024

Revision notes

Manuscript ID: PONE-D-23-34793R1

Title: Life tables in entomology: a discussion on tables’ parameters and the importance of raw data.

Journal: PLOS ONE

Decision letter and response to the Editor

Dear Dr. Rossini,

Thank you for submitting your manuscript to PLOS ONE. After careful consideration, we feel that it has merit but does not fully meet PLOS ONE’s publication criteria as it currently stands. Therefore, we invite you to submit a revised version of the manuscript that addresses the points raised during the review process.

We look forward to receiving your revised manuscript.

Kind regards,

Ramzi Mansour

Academic Editor

PLOS ONE

Journal Requirements:

Response:

Dear Dr. Mansour,

Thank you very much for your time and for the continued interest in our manuscript PONE-D-23-34793R1 entitled “Life tables in entomology: a discussion on tables’ parameters and the importance of raw data” for publication in PLOS ONE after a minor revision.

We sincerely appreciated all the positive comments and suggestions provided by the Reviewers and a point-by-point response to all the questions is provided below this document. During the revision we carefully addressed all the suggestions, with the hope to have sufficiently increased the quality of the manuscript. We renew our availability for any further question or request, if needed. Thank you again for considering our manuscript.

Sincerely,

Luca Rossini, on behalf of the co-authors.

Reviewers' comments

Reviewer 1

Reviewer 1: The authors have adequately addressed my comments and (in my opinion) those of the other reviewers, and the paper can now be accepted in its revised form. I commend them on putting together a great piece of work - I'm sure this will be an important paper in the field.

Response: Dear Reviewer 1, thank you once again for your time dedicated to revise our manuscript. We are sincerely grateful for the very positive comment, it means much to us. We hope to reach soon the next steps in this important research, that we believe is very important to improve the data collection and sharing in Entomology, Ecology, and Biological Sciences at large.

Thank you again for helping us to improve this manuscript and for the very helpful suggestions provided during the revision.

Reviewer 2

Reviewer 2: Authors made a great effort to address all reviewers’ comment and incorporate changes in their article, which I believe are a significant improvement. I only have very minor comments below.

Response: Dear Reviewer 2, thank you once again for your time dedicated to revise our manuscript, and for the helpful comments and suggestions provided during the rounds of review. We sincerely appreciated the very positive comment about our study, and we hope to have reached, during this revision, a form that is acceptable for publication. A point-by-point response to all the questions follows below this document, and we hope that our changes fits with your expectations. We also renew our availability for any further change or request, if needed.

Thank you again for helping us to improve this manuscript and for the very helpful suggestions provided during the revision.

Reviewer 2: L25 “the benefits”

Response: Thank you very much for this suggestion. We have corrected the text accordingly.

Reviewer 2: L30 “biological aspects”.

Response: Thank you for this suggestion. We have corrected the text accordingly.

Reviewer 2: L31 avoid using “evolution” in a demographic sense. I recommend “changes in the population”, or even “demographic changes”.

Response: Thank you for this suggestion. We have changed the text using the suggested alternative “demographic changes”.

Reviewer 2: L31 “highlights this by” or “highlights this aspect by” sounded better

Response: Thank you for this suggestion. We have corrected the text accordingly.

Reviewer 2: L46 “the individuals in a population” or “from a population”

Response: Thank you for this suggestion. We corrected the text accordingly.

Reviewer 2: L49 why did you remove “a” before cohort ? Are you sure this is correct English in current form?

Response: Thank you for this comment. We have restored the “a” before “cohort”.

Reviewer 2: L54 “to obtain various information”

Response: Thank you for this suggestion. We have changed the text accordingly.

Reviewer 2: L56 “The age-stage distribution describes… that compose the insect’s life cycle.”

Response: Thank you for this comment. We have changed the text accordingly.

Reviewer 2: L76 “outside such thermal range”

Response: Thank you for this comment. We have modified the text accordingly.

Reviewer 2: L123 “this is not yet common practice, which produces a loss”

Response: Thank you for this suggestion. We have modified this part of the text accordingly.

Reviewer 2: L124 “Remarkably, the unavailability of data”. A more offline comment here: a lot of papers publish a statement similar to “data are available from authors upon request”, notably because publishing data still remains a quite tedious, non straightforward and sometimes slow process, but most importantly also costly, and the community lacks general guidelines of the multiple ways to publish data. First, while this requires lots of efforts to collect data a posteriori from authors, I wonder how often authors do respond to these post-publication queries, is it something that the authors of the present paper have a sense of ? Second, wouldn’t it also be the role of publishers and journals to build up tools and guidelines about the so many available ways to publish data beyond the gold standards (e.g., Dryad), how are all the alternative (free github repositories, as supplementary materials of a publication, etc.) valid.

Response: Thank you for this very interesting comment. We totally agree with you, and it is an aspect that we wanted to highlight with this paper as well. Data sharing in many fields of Life Sciences, as Entomology or Ecology, is strongly limited by the lack of a standard for the data collection and sharing. This work is a starting point for a common goal that we should achieve all together as a community. We should talk more about the problem of data sharing, and we should propose solutions in order to reach a standard, as already carried out in other fields of research. Presenting the problem is of course the first step, but much work is still needed, and we hope that the readers will support our long-term mission. Of course, Journals can enhance the data sharing, but before we need a standard. Having a common database to share entomological data would be the best option, but also if the data are published on public and free repositories (such as GitHub) following a given standard guideline would be a good option. Well, we have still much work to do and many challenges to cope, but we are sincerely glad to know that there are other researchers that have our same feelings. We found these two rounds of review very constructive not only for the feedback received on the manuscript content, but also for the personal opinions of the Reviewers, including yours, that are a source of inspiration for the future. To answer the question, we do not have a clear idea of how frequently the corresponding authors respond to requests of data sharing, but we think that it depends on the research groups. An alternative statement that we often find in many published papers is “The data are available from the authors under reasonable request”: well, what is “reasonable” or not is very hard to understand..!

Reviewer 2: L135 does it really assume “identical”, or instead “interchangeable” as random draws of a single population? A random draw does not assume identity, but representativeness. The distribution will be imperfect (unrealistic) in case the draw is too small relative to the size of the population, and in case the population itself is not representative of the species. I’m not very sure I agree with the following argument L135-137; bootstrapping will smooth-out the distribution and a distribution is always based on individuals’ traits, this last sentence as currently written seems to oppose bootstrapping to individual-based data, so I don’t understand…

Response: Thank you very much for this comment. We have modified the lines 135-137 accordingly, to make this part of the text clearer. The bootstrap technique hides the actual distribution of the development times, this was the meaning behind this part of the text. We hope that now the sentence is clearer and that our change fits with your expectations.

Reviewer 2: L147 “of insect population models”, I assume not all modellers deal with pests.

Response: Thank you for this suggestion. We have changed the text accordingly.

Reviewer 2: L148 “exclusive usage of life tables’ synthetic information”

Response: Thank you for this suggestion. We have modified the text accordingly.

Reviewer 2: L200 is it instead “implies that individuals from a single population can have two different developmental times” ?

Response: Thank you for this suggestion. Yes, the meaning is correct, and the sentence was modified accordingly.

Reviewer 2: L316 “are both publicly”

Response: Thank you for this suggestion. We have modified this sentence accordingly.

Reviewer 2: L369 “Furthermore, any synthetic parameters from descriptive statistics will always come with some loss of information”.

Response: Thank you for this suggestion. We have modified this sentence accordingly.

---

## [Editor Report · Decision Letter 2]

13 Feb 2024

Life tables in entomology: a discussion on tables' parameters and the importance of raw data.

PONE-D-23-34793R2

Dear Dr. Rossini,

We’re pleased to inform you that your manuscript has been judged scientifically suitable for publication and will be formally accepted for publication once it meets all outstanding technical requirements.

Kind regards,

Ramzi Mansour

Academic Editor

PLOS ONE

---

## [Editor Report · Acceptance letter]

26 Feb 2024

PONE-D-23-34793R2 

PLOS ONE

Dear Dr. Rossini, 

I'm pleased to inform you that your manuscript has been deemed suitable for publication in PLOS ONE. Congratulations! Your manuscript is now being handed over to our production team.

Kind regards, 

on behalf of

Dr. Ramzi Mansour 

Academic Editor

PLOS ONE